

# Emergent constraints on Equilibrium Climate Sensitivity in CMIP5: do they hold for CMIP6?

Manuel Schlund[1], Axel Lauer[1], Pierre Gentine[2,3], Steven C. Sherwood[4], Veronika Eyring[1,5]

[1]Deutsches Zentrum für Luft- und Raumfahrt e.V. (DLR), Institut für Physik der Atmosphäre, Oberpfaffenhofen, Germany.

[2]Department of Earth and Environmental Engineering, Columbia University, New York, NY 10027.

[3]Earth Institute and Data Science Institute, Columbia University, New York, NY 10027.

[4]Climate Change Research Centre and ARC Centre of Excellence for Climate System Science, University of New South Wales, Sydney 2052, Australia.

[5]University of Bremen, Institute of Environmental Physics (IUP), Bremen, Germany.

*Correspondence to*: Manuel Schlund (manuel.schlund@dlr.de)

**Abstract.** An important metric for temperature projections is the equilibrium climate sensitivity (ECS) which is defined as the global mean surface air temperature change caused by a doubling of the atmospheric $CO_2$ concentration. The range for ECS assessed by the Intergovernmental Panel on Climate Change (IPCC) Fifth Assessment Report is between 1.5 and 4.5 K and has not decreased over the last decades. Among other methods, emergent constraints are potentially promising

approaches to reduce the range of ECS by combining observations and output from Earth System Models (ESMs). In this study, we systematically analyze 11 published emergent constraints on ECS that have mostly been derived from models participating in the Coupled Model Intercomparison Project Phase 5 (CMIP5) project. These emergent constraints are – except for one that is based on temperature variability – all directly or indirectly based on cloud processes, which stem the major source of uncertainty in ECS. The focus of the study is on testing if these emergent constraints hold for ESMs

participating in the new Phase 6 (CMIP6). Since none of the emergent constraints considered here has been derived using the CMIP6 ensemble, CMIP6 can be used for cross-checking of the emergent constraints on a new model ensemble. The application of the emergent constraints to CMIP6 data shows a large decrease of the correlation coefficients for most emergent relationships, indicating that nearly all of the constraints are less skillful in predicting ECS than they were in CMIP5. Many do not appear sufficiently skillful to be useful in constraining ECS, and several of them do not pass a

significance test. This is likely because of changes in the representation of cloud processes from CMIP5 to CMIP6, but may in some cases also be due to spurious statistical relationships or a too small number of models in the ensemble the emergent constraint was originally derived from. The emergent-constrained best estimate of ECS increased from CMIP5 to CMIP6, with a best estimate range of 2.97 – 3.88 K for CMIP5 and 3.41 – 4.36 K for CMIP6. This can be at least partly explained by the increased number of high-ECS models in CMIP6 with an ECS above 4.5 K without a corresponding change in the



constraint predictors. Our results support previous studies concluding that emergent constraints should be based on an independently verifiable physical mechanism, and that emergent constraints focusing on specific processes contributing to ECS are more promising than emergent constraints based on statistical model analysis.

## 1    Introduction

A bulk measure of the sensitivity of the climate system to carbon dioxide in the atmosphere ($CO_2$) is commonly expressed by
the equilibrium climate sensitivity (ECS), an idealized metric defined as the mean global surface air temperature change that results from a doubling of the atmospheric $CO_2$ concentration once the climate system reached equilibrium. In 1979, the Charney report determined an ECS range of 1.5 to 4.5 K for the Earth system (Charney et al., 1979). This range has not changed substantially over time (Meehl et al., 2020). In the Intergovernmental Panel on Climate Change (IPCC) Fifth Assessment Report (AR5), the reported likely range of ECS based on multiple lines of evidence is still between 1.5 and 4.5
K, whereas ECS calculated from climate and Earth system models participating in the Coupled Model Intercomparison Project Phase 5 (CMIP5, Taylor et al. (2012)) supporting AR5 is ranging between 2.1 and 4.7 K (Collins et al., 2013).

This large range of global climate sensitivity values can be largely attributed to differences in cloud feedbacks (Boucher et al., 2013). In particular, model differences in the change in shortwave reflection of low-level clouds changes in response to climate change dominate the uncertainties in the global warming projections, particularly in the tropics but also in mid-
latitudes (Brient and Schneider, 2016; Vial et al., 2013). Over the years, various lines of evidence have been exploited to constrain the range of ECS, including paleoclimate data and analysis of the current observed warming trend (Knutti et al., 2017a).

The use of emergent constraints is another promising approach to reduce the uncertainty in ECS (Eyring et al., 2019). Originally applied to the hydrological cycle and the snow-albedo feedback (Allen and Ingram, 2002; Hall and Qu, 2006),
emergent constraints offer the possibility to constrain future projections of Earth system model (ESM) ensembles with observations. Their theoretical basis is an emergent relationship between an observable quantity in the past or present-day climate and a quantity related to the future climate (such as for example ECS). Typically, the observable quantity is related to a climate feedback allowing the emergent relationship to be physically motivated by some key processes driving this feedback. Such a physical mechanism is a crucial prerequisite for the plausibility of an emergent constraint: due to large
number of possible observables and small number of models, spurious emergent relationships are possible just by chance, which was shown by statistical tests (Caldwell et al., 2014). Caldwell et al. (2018) evaluated the credibility of several published emergent constraints on ECS. Using a feedback decomposition analysis, they assessed whether the published emergent relationship could be explained by the proposed mechanism. Out of 19 emergent constraints on ECS, only four of them were considered credible, while the rest of them were considered as either not plausible or could not be tested using
this approach.



In this paper, we analyze 11 published emergent constraints on ECS which are summarized in Table 1 and assess whether they still hold for the new CMIP6 model ensemble (Eyring et al., 2016). We first calculate these emergent constraints for the ensemble that was used to derive the majority of them – CMIP5 (Taylor et al., 2012) – and then test whether they still hold in the CMIP6 ensemble. The only exception here is an emergent constraint by Volodin (2008) that was derived on CMIP3 data.

While the model range of ECS in CMIP5 is between 2.1 to 4.7 K, the CMIP6 model range is larger (1.8 to 5.6 K) with some models showing ECS values above 5 K (Meehl et al., 2020), see Figure 1. Possible reasons for this increased ECS range are changes in the extratropical cloud parametrizations and microphysics in the CMIP6 models (Zelinka et al., 2020). However, despite including more cloud physical processes, further analyses suggest that the high sensitivity models might overestimate the future warming trend (Forster et al., 2020; Tokarska et al., 2020). The large ECS range in CMIP6 emphasizes the need

for reliable methods to constrain the uncertainty range of future climate projections with observations. The CMIP6 ensemble can therefore be used for an independent testing of the constraints on previously unknown data. If the proposed underlying physical mechanisms are robust, the emergent constraints would be expected to hold when applied to CMIP6 data. In this analysis we thus test the robustness of the constraints to new models and models with advances in model design over time. In addition, we test the statistical significance of the relationships found in either model ensemble using a bootstrap resampling

approach.

Section 2 provides an overview of the data and methods used. Section 3 gives a discussion of the 11 emergent constraints on ECS and their results when derived from the CMIP5 or CMIP6 ensemble including a detailed analysis of their statistical significance. The paper ends with a discussion and summary in sections 4 and 5.

## 2    Data and methods

### 2.1    Equilibrium Climate Sensitivity (ECS)


In this study we use the output from climate models participating in CMIP5 and CMIP6, see Table 2 and Table 3, respectively. Since running a fully-coupled ESM into equilibrium is computationally expensive (this would require thousands of model years, see Rugenstein et al. (2020)), ECS is typically approximated by a so-called "effective climate sensitivity". This quantity is calculated by applying the Gregory regression method (Gregory et al., 2004), based on an

assumed relationship between forcing and response

$$N = \Delta F + \lambda \Delta T \qquad \textbf{(1)}$$

with the top-of-atmosphere (TOA) net downward radiation $N$ equal to the heat storage flux in the system, the external greenhouse gas (here $CO_2$ only) forcing $\Delta F$ and a global mean surface air temperature change $\Delta T$ with $\lambda$ the feedback parameter. This equation forms a linear regression model for $N(\Delta T)$ with slope $\lambda$ and intercept $\Delta F$. If these regression parameters are calculated from annual means of the quantities $N$ and $\Delta T$ from a 150-year simulation where the atmospheric



$CO_2$ concentration is abruptly quadrupled relative to a pre-industrial simulation and assume radiative equilibrium i.e. negligible heat storage flux ($N = 0$), the ECS can be evaluated as

$$ECS = -\frac{F}{2\lambda}. \tag{2}$$

The factor 2 in the denominator accounts for the fact that we use an abrupt four times $CO_2$ simulation, whereas ECS is defined for a doubling of the atmospheric $CO_2$ concentration.

## 2.2    Calculation of emergent constraints on ECS

An overview of the 11 emergent constraints analyzed in this study including the variables required for their calculations is given in Table 1 and the following section. In all cases, we use the historical simulations of CMIP5 and CMIP6 in order to ensure maximum agreement with the observational data. If necessary, the historical simulation of CMIP5 is extended after its final year 2005 with data from the RCP 8.5 scenario (Riahi et al., 2011). Note that we only use data through 2014, during which time all RCP scenarios behave similarly and the choice of the scenario is not expected to affect results considerably.

Such an extension is not needed for CMIP6 models as their historical simulations cover a longer time period until 2014.

To evaluate the resulting constrained probability distribution of ECS, we use the following nomenclature: let $x_m$ be the x-axis variable (i.e. the observable, constraining variable) of climate model $m$ and $y_m$ its corresponding target variable (ECS in our case). Following Cox et al. (2018), we use ordinary least squares regression to fit the linear model

$$\hat{y}_m(x) = a + bx_m, \tag{3}$$

where $\hat{y}_m$ is the predicted target variable for predictor $x_m$, $a$ the intercept of the linear regression line and $b$ the slope of the

linear regression line. Fitting the regression model includes minimizing the standard error s of the estimate

$$s^2 = \frac{1}{M-2} \sum_{m=1}^{M} (y_m - \hat{y}_m)^2, \tag{4}$$

where $M$ is the total number of climate models.

In the standard emergent constraint approach, the constrained best estimate for the target variable $y$ (here ECS) is given by the regression $\hat{y}(x_0)$ evaluated at an observed or observationally based (in case of using reanalysis data) value $x_0$. In that case, the uncertainty in predicting that best estimate is given by the standard prediction error

$$\sigma_{\hat{y}}^2(x_0) = s^2 \left( 1 + \frac{1}{M} + \frac{(x_0 - \bar{x})^2}{\sum_{m=1}^{M}(x_m - \bar{x})^2} \right). \tag{5}$$

Here, $\bar{x}$ is the arithmetic mean of $x$ over all models. Assuming Gaussian errors, this equation defines the conditional probability density function (PDF) for predicting a value of $y$ given $x_0$, i.e. the posterior distribution:

$$P(y|x_0) = \frac{1}{\sqrt{2\pi\sigma_{\hat{y}}^2(x_0)}} \exp\left( -\frac{(y - \hat{y}(x_0))^2}{2\sigma_{\hat{y}}^2(x_0)} \right). \tag{6}$$



This distribution can be interpreted as the posterior distribution in the regression model based on climate model output but constrained by matching the observation $x_0$. However, the observation of $x_0$ is not error free and has uncertainties associated with it. Assuming again Gaussian uncertainties, the resulting probability density for $x$ given the observation $x_0$ is given by

$$P(x) = \frac{1}{\sqrt{2\pi\sigma_{x_0}^2}}\exp\left(-\frac{(x-x_0)^2}{2\sigma_{x_0}^2}\right),$$

(7)

where $x_0$ is the best estimate because the error is assumed unbiased, and $\sigma_{x_0}^2$ the variance of the observation about the true value. In a final step, numerical integration is used to calculate the marginal probability density for the constrained prediction of the target variable $y$:

$$P(y) = \int_{-\infty}^{+\infty} P(y|x)P(x)dx$$

(8)

In assigning probabilities via equation (6) we have assumed that $P(y|y_0) = P(y_0|y)$ ($y_0$ is the unknown true value of $y$) and thereby have implicitly assumed a uniform prior probability density in ECS – in other words, that an ECS near 8 K is
just as probable as one near 4 K if both are equally consistent with the observation $x_0$. We do this for simplicity. Note that if we had instead applied this procedure to the climate radiative feedback parameter, which is arguably more physical than applying it to ECS (e.g. Roe and Baker (2007)), the resulting ECS PDFs would have non-symmetric shapes and different means. The conclusions of this study regarding changes from CMIP5 to CMIP6 would be unchanged however in either case. Even though it is also possible to use previous information (e.g., the range seen in GCMs) to inform a prior resulting in a
narrower posterior PDF with the emergent constraint added, here we focus on the constraining power of emergent constraints on their own. To assess the effectiveness of a constraint, the emergent-constrained PDFs are compared with published ECS ranges based on expert judgements and other information: if the constraint-based PDF is wider than the one from previous estimates, the constraint brings little added value, even if its validity has been shown.

### 2.3    Statistical significance of emergent constraints

In this study, we evaluate the statistical significance of the different emergent constraints on ECS. The term "statistical significance" refers to the sensitivity of the regression model to changes in the input data, i.e. the removal or addition of datasets. The basis for this evaluation is a non-parametric bootstrapping approach (similar to the one used by Zhai et al. (2015)). For this, we generate 100,000 bootstrap samples of size $M$ for every emergent constraint by randomly drawing from the original sample $\{(x_m, y_m)\}$ with replacement ($M$ is the total number of climate models for which all data required for the
considered emergent constraint are available). For each bootstrap sample, the linear (Pearson) correlation coefficient $r$ is calculated. Using the probability distribution of the bootstrap samples of $r$, we define a $p$-value as the probability that $r$ exhibits the opposite sign as originally expected from the emergent relationship (see shaded areas in the right columns of Figure 2 to Figure 5). In other words, $p := \text{CDF}(0)$ ($p := 1 - \text{CDF}(0)$) for an expected positive (negative) emergent relationship where CDF refers to the cumulative distribution function of the bootstrap samples of $r$. In this context we



introduce the following nomenclature: an emergent relationship is called "highly significant" if $p < 0.02$, "barely significant" if $0.02 \leq p < 0.05$, "almost significant" if $0.05 \leq p < 0.1$, and "far from significant" if $p \geq 0.1$.

### 2.4    ESMValTool

All figures in this paper are produced with the Earth System Model Evaluation Tool (ESMValTool) version 2.0 (v2.0) (Eyring et al., 2020; Lauer et al., 2020; Righi et al., 2020). The ESMValTool is an open-source community diagnostics and

performance metrics tool for the evaluation of Earth system models (https://www.esmvaltool.org/). An ESMValTool recipe (configuration file defining input data, preprocessing steps and diagnostics to be applied) is available that can be used to reproduce all figures in this paper. This also allows redoing the analysis presented in this study once new model simulations from CMIP6 or other model ensembles become available.

### 3    Comparison of emergent constraints on ECS for CMIP5 and CMIP6

In this section we describe and discuss the 11 emergent constraints on ECS summarized in Table 1 using CMIP5 and CMIP6 data (sections 3.1 to 3.11) and provide a best estimate for ECS and statistical significance of the 11 emergent constraints in section 3.12. While most of these emergent constraints have been derived using data from the CMIP5 and/or CMIP3 ensembles, to our knowledge none of them has been evaluated on the CMIP6 ensemble so far. The results for the individual emergent constraints described in the following are shown in Figure 2 to Figure 5. Table 4 shows corresponding IPCC likely

ranges (i.e. 66% confidence interval) of ECS derived from the probability distributions given by equation (8) and the *p*-values used to assess the significance of the emergent relationships.

### 3.1    Response of shortwave cloud reflectivity to changes in sea surface temperature (BRI)

In this emergent constraint proposed by Brient and Schneider (2016), ECS is correlated with the tropical low-level cloud (TLC) albedo. Differences in the TLC albedo account for more than half of the variance of the ECS in the CMIP5 ensemble.

Following Brient and Schneider (2016), TLC regions are defined as grid points that are in the driest quartile of 500 hPa relative humidity of all grid cells over the ocean between 30°S and 30°N. The albedo of the TLC is obtained by calculating the ratio of TOA shortwave cloud radiative forcing and solar insolation averaged over the TLC region. The regression coefficients of deseasonalized variations of TLC shortwave albedo and sea surface temperature SST (in % per K) are then used as an emergent constraint for ECS. Here, we use observational data from HadISST for SST (Rayner et al., 2003), ERA-

Interim for 500 hPa relative humidity (Dee et al., 2011) and CERES-EBAF (Loeb et al., 2018) for the TOA radiative fluxes over the time period 2001–2005. With the exception of SST, where data from the Extended Reconstructed Sea Surface Temperature (ERSST) is used (Smith and Reynolds, 2003), the original publication used similar observation-based datasets. Our analysis yields a likely range for ECS of 3.72 K ± 0.56 K for CMIP5 ($R^2 = 0.38$) and 4.36 K ± 1.16 K for CMIP6, with much lower $R^2 = 0.15$. The original publication stated a best estimate of 4.0 K, with a very low likelihood of values below





2.3 K (90% confidence). The PDFs of the Pearson correlation coefficient $r$ obtained by the non-parametric bootstrapping approach (right column in Figure 2) show that the emergent relationship exhibits the expected negative correlation for CMIP5 and CMIP6. The emergent relationship is highly significant for CMIP5 ($p = 0.0002$) and barely significant for the CMIP6 ensemble ($p = 0.0219$).

### 3.2     Temperature variability (COX)

The emergent constraint on ECS proposed by Cox et al. (2018) uses interannual variation of global mean temperature calculated from its variance (in time) and one-year-lag autocorrelation. In contrast to the majority of emergent constraints which focus on cloud-related processes, this constraint is based on the fluctuation-dissipation theorem, which relates the long-term response of the climate system to an external forcing (ECS) to short-term variations of the climate system (climate variability). This arguably places the constraint on a more solid theoretical foundation, although several questions were
raised on the robustness of the results (Brown et al., 2018; Po-Chedley et al., 2018; Rypdal et al., 2018). As observational data, here we use the HadCRUT4 dataset (Morice et al., 2012) over the time period 1880-2014. For the COX constraint, we assess a likely ECS range of 3.03 K ± 0.71 K for CMIP5 ($R^2 = 0.31$) and 3.44 K ± 1.15 K for CMIP6 ($R^2 = 0.08$). Cox et al. (2018) derived a likely range of 2.8 K ± 0.6 K from a different subset of CMIP5 models. For CMIP6, the distribution of $r$ evaluated from the bootstrap samples (right column of Figure 2) shows high probability densities around $r = 0$, which
means that many bootstrap samples show a very low correlation coefficient. Moreover, while the majority of bootstrap samples indicate a positive $r$, there is also a considerably high fraction of bootstrap samples that show a negative correlation (orange shaded area). In contrast to that, the distribution of $r$ for the CMIP5 ensemble supports a clear positive correlation. Consequently, the COX emergent relationship is highly significant for the CMIP5 ensemble ($p = 0.0010$), but only almost significant for the CMIP6 ensemble ($p = 0.0545$).

### 3.3     Southern hemisphere Hadley cell extent (LIP)


The results of Lipat et al. (2017) show that the multi-year average extent of the Hadley cell correlates with ECS in CMIP5 models. The Hadley cell edge is defined as the latitude of the first two grid cells from the equator going south where the zonal average 500 hPa mass stream function calculated from December-January-February means of the meridional wind field changes sign from negative to positive. Lipat et al. (2017) explain this correlation by tying it to the observed correlation
of the interannual variability in mid-latitude clouds and their radiative effects with the poleward extent of the Hadley cell. For the calculation of the emergent constraint, we use reanalysis data from ERA-Interim (Dee et al., 2011) for the meridional wind speed over the time period 1980–2005. Our application of this emergent constraint gives ECS likely ranges of 2.97 K ± 0.76 K for CMIP5 ($R^2 = 0.18$) and 3.66 K ± 1.27 K for CMIP6 ($R^2 = 0.03$). The original publication does not specify an ECS range. The emergent constraint by Lipat et al. (2017) is highly significant for CMIP5 ($p = 0.0043$) but far from
significant for CMIP6 ($p = 0.2039$).





## 3.4 Large-scale lower-tropospheric mixing (SHD)

Sherwood et al. (2014) proposed that the degree of mixing in the lower troposphere determines the response of boundary-layer clouds and humidity to climate warming, as the associated moisture transport would increase rapidly in a warmer atmosphere due to the Clausius-Clapeyron relationship. The large-scale component $D$ of this mixing is defined as the ratio of

shallow to deep overturning. $D$ is calculated from the vertical velocities averaged over two height regions: 850 hPa and 700 hPa for shallow overturning and 600, 500 and 400 hPa for deep overturning. Both quantities are averaged over parts of the tropical ocean region away from the regions of highest SST and strongest mid-level ascent, in particular the region 160°W – 30°E, 30°S – 30°N, wherever air is ascending at low levels. As observationally based data, we use vertical velocities from ERA-Interim (Dee et al., 2011) over the time period 1989–1998 similar to the original publication. We derive ECS likely

ranges of 3.65 K ± 0.63 K for CMIP5 ($R^2 = 0.28$) and 3.74 K ± 1.11 K for CMIP6 ($R^2 = 0.04$). Sherwood et al. (2014) do not give a best estimate for ECS based on the large-scale component of mixing $D$ or its small-scale counterpart $S$ (section 3.5) but for the sum of $D+S$ only (see section 3.6). The evaluation of the bootstrap distribution of $r$ indicates that the SHD constraint is highly significant for the CMIP5 ensemble ($p = 0.0006$) but far from significant for the CMIP6 ensemble ($p = 0.1120$).

## 3.5 Small-scale lower-tropospheric mixing (SHS)


The small-scale mixing $S$ (Sherwood et al., 2014) is calculated from the differences in relative humidity and temperature between 700 and 850 hPa. The differences are averaged over all grid cells within the upper quartile of the annual mean 500 hPa ascent rate (within ascending regions) in the tropics. The tropics are defined as region between 30°S and 30°N. In the Cloud Feedback Model Intercomparison Project models (CFMIP, Webb et al. (2017)), for which convective tendencies were

available, upward moisture transport by parameterized convection was shown to increase more rapidly with warming for higher values of $S$. We use reanalysis data from ERA-Interim (Dee et al., 2011) for temperature and relative humidity to calculate the observationally based constraint (1989–1998). Our analysis shows a likely range of ECS of 3.07 K ± 0.68 K for CMIP5 ($R^2 = 0.13$) and 3.41 K ± 1.13 K for CMIP6 ($R^2 = 0.15$). The correlation of $S$ and ECS is almost significant for the CMIP5 ($p = 0.0581$) and the CMIP6 ($p = 0.0638$) ensemble. The SHS constraint is one of the two analyzed emergent

constraints (ZHA being the other exception) that shows a higher coefficient of determination $R^2$ for the CMIP6 than for the CMIP5 ensemble.

## 3.6 Lower tropospheric mixing index (SHL)

The lower tropospheric mixing index (LTMI) formulated by Sherwood et al. (2014) is defined as the sum of the small-scale mixing $S$ (see section 3.5) and the large-scale mixing $D$ (see section 3.4), which are supposed to capture complementary

components of the total mixing phenomenon. Sherwood et al. (2014) argue that the increase in dehydration depends on initial mixing linking it to cloud feedbacks and thus also to ECS. For this constraint, we derive an ECS likely range of 3.42 K





$\pm$ 0.63 K for CMIP5 ($R^2 = 0.41$) and 3.65 K $\pm$ 1.05 K for CMIP6 ($R^2 = 0.19$). Sherwood et al. (2014) give a best estimate of about 4 K with a lower limit of 3 K. As illustrated by the right column of Figure 3, the SHL emergent relationship is highly significant for both considered climate model ensembles, CMIP5 ($p = 0.0001$) and CMIP6 ($p = 0.0114$), although

weaker in the newer ensemble.

### 3.7 Error in vertical profile of relative humidity (SU)

Another emergent constraint on ECS that targets uncertainties in cloud feedbacks was proposed by Su et al. (2014). They show that changes in the Hadley circulation are physically connected to changes in tropical clouds and thus ECS. Consequently, the inter-model spread in the change of the Hadley circulation in an ensemble of climate models is well

correlated with the corresponding changes in the TOA cloud radiative effect. Moreover, Su et al. (2014) found a correlation between a model's ECS and its ability to represent the present-day Hadley circulation. The latter is calculated from the tropical (45°S – 40°N) zonal-mean vertical profiles of relative humidity from the surface to 100 hPa. These profiles are then used to define the x-axis of the SU constraint by calculating a performance metric based on the slope of the linear regression between a climate model's relative humidity profile and the corresponding observational reference. Similarly to the original

publication, we use humidity observations from AIRS (Aumann et al., 2003) for pressure levels greater than 300 hPa and MLS-Aura data (Beer, 2006) for pressure levels of less than 300 hPa. Our analysis yields a constrained likely range of ECS of 3.30 K $\pm$ 0.90 K for CMIP5 ($R^2 = 0.08$) and 3.69 K $\pm$ 1.59 K for CMIP6 ($R^2 = 0.03$). The original publication gives a best estimate of 4 K with a lower limit of 3 K. Figure 4 shows that in addition to the low $R^2$ values, the emergent relationship shows different slopes for CMIP5 and CMIP6. For the CMIP5, the expected positive correlation is found, while

for CMIP6, a negative correlation is found. This suggests that the constraint is not working (any more) when applied to the CMIP6 data. Consequently, the SU constraint is almost significant for the CMIP5 ensemble ($p = 0.0919$) and far from significant for the CMIP6 ensemble ($p = 0.8573$).

### 3.8 Tropical mid-tropospheric humidity asymmetry index (TIH)

Tian (2015) found a link between mid-tropospheric humidity over the tropical Pacific and simulated moisture, precipitation,

clouds, and large-scale circulation and thus ECS in CMIP3 and CMIP5 models. The study explains this link with the similarity of mid-tropospheric humidity and precipitation patterns as both are related to the ITCZ. The proposed tropical mid-tropospheric humidity asymmetry index to constrain ECS is defined as relative bias (in percent) in simulated annual mean 500 hPa specific humidity averaged over the Southern Hemisphere (SH) tropical Pacific (30°S – 0°, 120°E – 80°W) minus the bias averaged over the Northern Hemisphere (NH) tropical Pacific (20°N - 0°, 120°E - 80°W) when compared

with observations. Here, we use humidity observations from AIRS (Aumann et al., 2003) over the time period 2003–2005 as reference dataset. We assess a likely ECS range of 3.88 K $\pm$ 0.78 K for CMIP5 ($R^2 = 0.24$) and 4.07 K $\pm$ 1.21 K for CMIP6



($R^2 = 0.07$). Tian (2015) specifies a best estimate of 4.0 K. The emergent relationship is highly significant for the CMIP5 ensemble ($p = 0.0002$) and barely significant for the CMIP6 ensemble ($p = 0.0454$).

### 3.9 Southern ITCZ index (TII)

In addition to the humidity index, Tian (2015) proposed an emergent constraint on ECS based on the southern ITCZ index (Bellucci et al., 2010; Hirota et al., 2011). This index is defined as the climatological annual mean precipitation bias averaged over the south-eastern Pacific (30°S – 0°, 150°W – 100°W). The southern ITCZ index is calculated in mm day$^{-1}$ and dominated by the so-called double ITCZ, a common problem in many CMIP5 climate models. Tian (2015) found a link between double-ITCZ bias and simulated moisture, precipitation, clouds, and large-scale circulation in CMIP3 and CMIP5

models. He argues that this could explain the link found between the double-ITCZ bias and ECS. As reference data, we use observed precipitation data for the years 19862005 from GPCP (Adler et al., 2003). We calculate an ECS likely range of 3.87 K ± 0.70 K for CMIP5 ($R^2 = 0.33$) and 4.11 K ± 1.17 K for CMIP6 ($R^2 = 0.06$). Tian (2015) specifies a best estimate of 4.0 K. As shown by the right column of Figure 4, the TII emergent relationship is highly significant for the CMIP5 ensemble ($p = 0.0004$), but only almost significant for the CMIP6 ensemble ($p = 0.0634$).

### 3.10 Difference between tropical and mid-latitude cloud fraction (VOL)

The study by Volodin (2008) aims at the geographical distribution of clouds in climate models. It was the first published emergent constraint on ECS, relying on models from CMIP3, such that both CMIP5 and CMIP6 are out-of-sample tests for this constraint. He shows that high ECS models tend to simulate a higher total cloud cover over the southern mid-latitudes and a lower total cloud cover over the tropics (relative to the multi-model mean). This can be used to establish an emergent

relationship between the ECS and the difference in tropical total cloud cover (28°S – 28°N) and the southern mid-latitude total cloud cover (56°S – 36°S). Analogous to the original study, we use the ISCCP-D2 data (Rossow and Schiffer, 1991) as observational reference. For the VOL constraint, we calculate a constrained likely range of ECS of 3.74 K ± 0.64 K for CMIP5 ($R^2 = 0.38$) and 4.14 K ± 1.13 K for CMIP6 ($R^2 = 0.16$), whereas the original publication gives a range of 3.6 K ± 0.4 K (standard deviation) for a climate model ensemble of CMIP3 models. The emergent constraint by Volodin (2008) is

one of the two emergent constraints that is highly significant for both the CMIP5 ($p = 0.0004$) and the CMIP6 ($p = 0.0057$) ensemble.

### 3.11 Response of seasonal marine boundary layer cloud fraction to SST changes (ZHA)

Zhai et al. (2015) focus on the variations of marine boundary layer clouds (MBLC), which largely contribute to the shortwave cloud feedback and thus to the uncertainty in modeled ECS. Their central quantity is the response of the MBLC

fraction to changes in the sea surface temperature (SST) in subtropical oceanic subsidence regions for both hemispheres (20° – 40°). On short (seasonal) and long (centennial under a forcing) time scales, this quantity is well correlated with ECS





among an ensemble of CMIP3 and CMIP5 models. Together with observations of cloud fraction from CloudSat/CALIPSO (Mace et al., 2009), SST from AMSRE SST (AMSR-E, 2011) and vertical velocity from ERA-Interim (Dee et al., 2011), the seasonal response of MBLC fraction to changes in SST forms an emergent constraint on ECS. We assess a likely ECS range
of 3.35 K ± 0.72 K for CMIP5 ($R^2 = 0.05$) and 3.81 K ± 0.60 K for CMIP6 ($R^2 = 0.61$). In their original publication, Zhai et al. (2015) found an ECS range of 3.90 K ± 0.45 K (standard deviation) for a combination of CMIP3 and CMIP5 models. In terms of statistical significance, the results of the ZHA constraints are somewhat surprising: although CMIP5 data (in combination with CMIP3 data) were successfully used in their original publication, our approach finds that the emergent relationship for the CMIP5 models is far from statistically significant. The reason for this disagreement is the set of climate
models used. For our analysis, we use 11 additional CMIP5 models that were not used in the original publication (i.e. ACCESS1-0, ACCESS1-3, bcc-csm1-1, bcc-csm1-1-m, CCSM4, GFDL-ESM2G, GFDL-ESM2M, IPSL-CM5A-MR, IPSL-CM5B-LR, MPI-ESM-MR and MPI-ESM-P). Due to a lack of publicly available data, the model CESM1-CAM5 that is used in the original publication is not included in our analysis. The effect of choosing different subsets of CMIP5 models on the emergent relationship is illustrated in Figure 6. Using the original CMIP5 models from the original publication gives a
considerably higher correlation ($R^2 = 0.38$) than using all available CMIP5 models ($R^2 = 0.05$). This result shows a strong dependency of this emergent constraint on the subset of climate models used. Moreover, strangely and uniquely among the metrics examined here, the ZHA constraint is highly significant for the CMIP6 ensemble ($p = 0.0022$) but far from significant in the updated CMIP5 ensemble ($p = 0.1195$).

### 3.12    Best estimates for ECS and statistical significance of the 11 emergent constraints

In most cases, the emergent relationships (left columns of Figure 2 to Figure 5) show the same sign of the slope for CMIP5 and CMIP6, with the SU constraint being the only exception. However, the coefficient of determination ($R^2$) is considerably lower for CMIP6 compared to CMIP5 for all but two constraints: SHS and ZHA. The probability distributions of the constrained ECS that we obtain (middle columns of Figure 2 to Figure 5) give similar results: except for the ZHA constraint, the constraint on the CMIP6 ensemble is weaker, i.e. the constrained PDFs derived from the CMIP6 ensemble are broader
than their respective CMIP5 counterparts. As shown in Table 4, for CMIP5, the range of the best estimates for ECS is 2.97 K to 3.88 K, while the corresponding CMIP6 best estimates are higher for each of the tested emergent constraints, resulting in an ECS range of 3.41 K to 4.36 K.

The $R^2$ of the emergent relationships and the constrained range of ECS each show a strong dependency on the climate model ensemble used, even though a physical explanation is given for each emergent constraint that is thought to be valid for every
climate model ensemble describing the real world. In order to assess changes in the skill of the emergent constraints when moving from CMIP5 to CMIP6 we use the degree of statistical significance relative to a standard $p = 0.05$ threshold ("highly", "barely", "almost", "far from") of the different emergent constraints shown in the right columns of Figure 2 to Figure 5 as a proxy. We consider reductions in the skill of the constraint as significant if the interquartile ranges of the bootstrapped correlation coefficient $r$ for CMIP5 and CMIP6 data do not overlap (colored boxes in Figure 7). This failure to



overlap is seen for all emergent constraints but SHS, confirming that most of the constraints have lost more skill than could be explained, by sampling uncertainty alone, if the models were independent. Only two constraints (SHL and VOL) show skill with high significance for both the CMIP5 and CMIP6 ensembles. Two more emergent constraints (BRI and TIH) are highly significant for CMIP5 but barely significant for CMIP6. Three other constraints are far from significant on CMIP6 (LIP, SHD and SU), but only one fails a significance test on CMIP5 (ZHA). Figure 7 gives an overview showing boxplots of the distributions for each of the 11 emergent constraints. Following the definition given in section 2.3, an emergent relationship is either highly or barely significant if the whiskers spanning the one-sided 95% confidence interval do not cross the horizontal line indicating $r = 0$. Emergent relationships with whiskers crossing that line are either almost significant or far from significant. For CMIP5, eight constraints are either highly or barely significant (BRI, COX, LIP, SHD, SHL, TIH, TII and VOL), while for CMIP6, only five constraints are either highly or barely significant (BRI, SHL, TIH, VOL and ZHA). Note that for better comparability, the signs of the correlation coefficients $r$ from the emergent relationships with expected negative slopes have been changed in Figure 7. Thus, positive values of $r$ in the figure indicate that the correlation coefficient matches the sign of the expected (published) relationship, while negative values indicate that the correlation coefficient does not match the expected sign. The latter is only the case for the SU constraint evaluated on the CMIP6 ensemble.

## 4 Discussion

As shown in the previous sections, most emergent relationships show smaller coefficients of determination when evaluated on the new CMIP6 ensemble compared to the CMIP5 ensemble. In this section, we discuss possible reasons for these differences. As reported by Caldwell et al. (2014), the large amount of data provided by modern ESMs can generate spurious correlations of variables between past climate and ECS just by chance, especially when only a small number of climate models is considered. This would cause the performance of the emergent constraint to be reduced on out-of-sample data (like the new CMIP6 ensemble), since the emergent relationship appeared just by chance and not because of a physically based mechanism.

A further reason for the weaker emergent relationships in CMIP6 may be the increased complexity of the participating ESMs. Each emergent constraint approach is based on the assumption that a single observable process or physical aspect in the current climate dominates the uncertainty in ECS. Some emergent constraints such as ZHA and BRI relate changes in cloud properties (here: low-level cloud fraction and cloud reflectivity) on seasonal or interannual time-scales (here: driven by changes in SST) to ECS. This means that it has to be implicitly assumed that the observable changes in these properties on seasonal or interannual time-scales are basically driven by the same mechanisms as the changes in cloud properties as a result of climate forcing. While this assumption seems to make sense, we do not know whether it actually holds in a significantly different climate or if other or additional mechanisms also might become important. For this reason it also remains unclear if the regions or cloud regimes that have been selected based on present-day climate and that are used to





calculate the emergent constraints will be equally important under significant climate change. For example Lauer et al. (2010) showed with a regional climate model that the relationship between cloud amount and lower tropospheric stability in the stratocumulus deck over the Southeast Pacific derived from present-day data will be altered under global warming.

For CMIP6 models, Zelinka et al. (2020) showed that cloud feedbacks and thus ECS in high-sensitivity models are dominated by changes in clouds over the Southern Ocean, while in CMIP3 and CMIP5 the uncertainty in cloud feedbacks is dominated by clouds in the subtropical subsidence regions. One might speculate that a possible reason for this might be an improved simulation of clouds over the Southern Ocean in some models (Bodas-Salcedo et al., 2019; Gettelman et al., 2019a) as shown for some pre-CMIP6 model versions evaluated by Lauer et al. (2018). The findings of Zelinka et al. (2020)

could also at least partly explain the larger inter-model spread in climate sensitivity due to more and different regions / clouds types dominating the cloud feedbacks resulting in a weaker emergent constraint compared with CMIP5 models. They found that on average, the shortwave low cloud feedback is larger in CMIP6 than in CMIP5, which they primarily relate to changes in the representation of clouds. As a possible explanation, Zelinka et al. (2020) give an increase in mean-state supercooled liquid water (i.e. increase in the cloud water liquid fraction) in mixed-phase clouds resulting in less pronounced

increases in low-level cloud cover and water content with warmer SSTs particularly in mid-latitudes.

We note that also observational uncertainties can play a role as using different observational datasets for a given variable as a proxy for observational uncertainty might lead to different emergent constraints. As this study uses only one combination of observational dataset(s) to calculate the emergent constraints as in the original published emergent constraint studies, the error estimations given by our analysis are expected to underestimate the true error. This could be investigated by systematic

tests using different observational datasets and/or combinations of thereof as a proxy for observational uncertainty. Where available, additionally observational uncertainty estimates could be used to give better estimates of the likely constrained range of ECS. A major challenge associated with this is, however, to determine how observational uncertainties propagate to the space-time scales represented by the models because of the typically not well known correlation of observational errors in space and time (e.g. Bellprat et al. (2017)).

**5    Summary**

This paper assesses 11 different emergent constraints on ECS, of which most are directly or indirectly related to cloud feedbacks, by applying them to results from ESMs contributing to CMIP5 and CMIP6. Of particular interest are the results from CMIP6, since all analyzed emergent constraints were published prior to the availability of CMIP6 data. In summary, we assess a range of 2.97 K to 3.88 K for the best estimates of ECS for CMIP5 and a range of 3.41 K to 4.36 K for CMIP6.

This increase in the best estimate of ECS can be found for every constraint that we analyzed, and can be at least partly explained by the increased multi-model mean ECS of CMIP6 which was not accompanied by systematic changes in the constraint variables that could explain this increase – leading to regression fits with higher intercept values at observed constraint values. This is also illustrated by the CMIP5 and CMIP6 multi-model means in the left columns of Figure 2 to



Figure 5 (colored dots), in which the connecting line between the CMIP5 and CMIP6 multi-model mean is not parallel to the
CMIP5 emergent relationships for all emergent constraints. However, these results need to be treated with great care as the
analysis showed that all considered emergent relationships are sensitive to outliers and the subset of the climate model
ensemble used to fit the emergent relationship. Moreover, our results also show that except for ZHA and SHS, all emergent
relationships are weaker (in terms of the coefficient of determination $R^2$) in CMIP6 compared to CMIP5, which means that
the corresponding emergent relationships are able to explain less of the ECS variation simulated by the newer CMIP6
models than by those of CMIP5.

Of the 11 emergent constraints analyzed, four are found to be "working" in the sense that they show statistically significant
skill on both the CMIP5 and CMIP6 ensembles: BRI, SHL, TIH and VOL. In contrast, the three emergent constraints LIP,
SHD and SU are found to be "not working anymore" as their $p$-values are well above 0.1 in CMIP6 (far from significant).
COX, TII and SHS are somewhat in-between and could be grouped as "indeterminate" as their $p$-values in CMIP6 dropped
from highly or barely significant to almost significant. It is noteworthy that among the group "working" three out of the four
emergent constraints point to rather high ECS values of above 4 K in CMIP6, while among the group "not working
anymore" two out of three emergent constraints point to rather small ECS values of 3.3 K and less. This might be evidence
that emergent constraints on ECS might point to rather higher than lower values in CMIP6.

Typically, studies proposing a single emergent constraint on ECS do not explicitly take into account model interdependency
and all approaches discussed above apply a linear regression of some kind to the model data. This means that it is implicitly
assumed that the individual data points (i.e. climate models) are independent. As some modeling groups provide output from
multiple ESMs and some ESMs from different modeling groups share components and code, this is clearly not the case.
Duplicated code in multiple models is expected to lead to an overestimation of the sample size of a model ensemble and may
result in spurious correlations (Sanderson et al., 2015). This limitation also applies to this study as the tests for significance
assume that all models are independent. Possible approaches could be to stop treating all models equally by either applying a
model weighting based on a model's interdependence with the other models or by simply reducing the ensemble size taking
into account models only that are above a given (yet to be defined) interdependence score. Promising approaches to quantify
the model interdependency that could be followed include, for example, the studies of Sanderson et al. (2015); Sanderson et
al. (2017) and Knutti et al. (2017b).

A further limitation of this study involves the calculation of significance for the different climate model ensembles: our non-
parametric bootstrapping approach calculates the spread of sample values one might obtain if the truth looked like the
sample, not the spread of true values consistent with the sample. This is particularly relevant for datasets with a large $R^2$ and
a small sample size, in which case the bootstrap spread will be overconfident. However, since most emergent constraints
show small $R^2$, this effect is expected to be small in our study. Also the calculation of the ECS itself is a source of
uncertainty: even though widely used in literature, the Gregory regression method (Gregory et al., 2004) is known to be only
an approximation of the true climate sensitivity. A recent paper (Rugenstein et al., 2020) shows that the true equilibrium
warming obtained from integrating the climate models until a new equilibrium is reached is 17% (median) higher than the



one estimated from the first 150 years of the simulation as done in the Gregory method. However, only a few ESMs provide simulations long enough to assess the true climate sensitivity. The CMIP endorsed LongRunMIP (Rugenstein et al., 2019)

could be a promising way to estimate the true climate sensitivity that can then be used to reevaluate emergent constraints and their proposed underlying physical mechanisms.

ECS is the product of the complex interactions of the many components. Thus, constraining ECS with a single physical process might overly simplify this problem. With increasing computational resources available to climate science, more and more detailed of these interactions can be taken into account in a modern ESM. In contrast, the predecessor versions CMIP3

and CMIP5 were less complex with fewer components, so constraining uncertainties of a single dominant process may have allowed for a more successful constraining of ECS than in more complex models. As a conclusion, we argue that to constrain ECS for the latest generation of climate models, it might be beneficial to apply multivariate approaches that are able to consider multiple (different) relevant physical processes at once and thus are able to get a broader picture of the complex reality. New machine learning techniques are a promising avenue forward for such multivariate approaches and for

constraining uncertainties in multi-model projections (Schlund et al., in review) with the aim of further improving climate modelling and analysis (Reichstein et al., 2019).



# 6   Tables

| Label | Reference | Short description of x-axis | Variables | Observations |
|---|---|---|---|---|
| BRI | (Brient and Schneider, 2016) | Response of shortwave cloud reflectivity to changes in sea surface temperature [% K$^{-1}$] | • Surface temperature ($ts$) <br> • Relative humidity ($hur$) <br> • Top-of-atmosphere (TOA) outgoing shortwave radiation ($rsut$) <br> • TOA outgoing shortwave radiation assuming clear sky ($rsutcs$) <br> • TOA incoming shortwave radiation ($rsdt$) | HadISST ($tos$) (Rayner et al., 2003), ERA-Interim ($hur$) (Dee et al., 2011), CERES-EBAF ($rsut$, $rsutcs$, $rsdt$) (Loeb et al., 2018) [2001-2005] |
| COX | (Cox et al., 2018) | Temperature variability metric [K] | • Surface air temperature ($tas$) | HadCRUT4 (Morice et al., 2012) [1880-2014] |
| LIP | (Lipat et al., 2017) | Southern hemisphere Hadley cell extent [°] | • Northward wind ($va$) | ERA-Interim (Dee et al., 2011) [1980-2005] |
| SHD | (Sherwood et al., 2014) | $D$ index (large-scale lower-tropospheric mixing) [1] | • Vertical velocity ($wap$) | ERA-Interim (Dee et al., 2011) [1989-1998] |
| SHL | (Sherwood et al., 2014) | Lower tropospheric mixing index (LTMI) [1] | • Relative humidity ($hur$) <br> • Air temperature ($ta$) <br> • Vertical velocity ($wap$) | ERA-Interim (Dee et al., 2011) [1989-1998] |
| SHS | (Sherwood et al., 2014) | $S$ index (small-scale lower-tropospheric mixing) [1] | • Relative humidity ($hur$) <br> • Air temperature ($ta$) <br> • Vertical velocity ($wap$) | ERA-Interim (Dee et al., 2011) [1989-1998] |
| SU | (Su et al., 2014) | Error in vertical profile of relative humidity [1] | • Relative humidity ($hur$) | AIRS (below 300hPa) (Aumann et al., 2003), MLS-Aura (above 300hPa) (Beer, |





| Label | Reference | Short description of x-axis | Variables | Observations |
|---|---|---|---|---|
| | | | | 2006) [2005-2010] |
| TIH | (Tian, 2015) | Tropical mid-tropospheric humidity index [%] | • Specific humidity (*hus*) | AIRS (Aumann et al., 2003) [2003-2005] |
| TII | (Tian, 2015) | Southern ITCZ index [mm day$^{-1}$] | • Precipitation (*pr*) | GPCP (Adler et al., 2003) [1986-2005] |
| VOL | (Volodin, 2008) | Difference in total cloud fraction between tropics (28°S – 28°N) and southern mid-latitudes (56°S – 36°S) [%] | • Total cloud area fraction (*clt*) | ISCCP D-2 (Rossow and Schiffer, 1991) [1980-2000]* |
| ZHA | (Zhai et al., 2015) | Response of seasonal marine boundary layer cloud fraction to change in sea surface temperature [% K$^{-1}$] | • Cloud area fraction (*cl*)<br>• Sea surface temperature (*tos*)<br>• Vertical velocity (*wap*) | CloudSat/CALIPSO (Mace et al., 2009), AMSRE SST (AMSR-E, 2011), ERA-Interim (Dee et al., 2011) [1980-2004]* |

**Table 1: Overview of the 11 emergent constraints on the ECS used in this study. Observations marked with an asterisk (*) are**
**identical with the ones used in original publication.**



| Index used in plots | Model | Reference |
|---|---|---|
| 1 | ACCESS1-0 | (Dix et al., 2013) |
| 2 | ACCESS1-3 | (Dix et al., 2013) |
| 3 | BNU-ESM | (Ji et al., 2014) |
| 4 | CCSM4 | (Gent et al., 2011; Meehl et al., 2012) |
| 5 | CNRM-CM5 | (Voldoire et al., 2013) |
| 6 | CNRM-CM5-2 | (Voldoire et al., 2013) |
| 7 | CSIRO-Mk3-6-0 | (Rotstayn et al., 2012) |
| 8 | CanESM2 | (Arora et al., 2011) |
| 9 | FGOALS-g2 | (Li et al., 2013) |
| 10 | GFDL-CM3 | (Donner et al., 2011) |
| 11 | GFDL-ESM2G | (Dunne et al., 2012) |
| 12 | GFDL-ESM2M | (Dunne et al., 2012) |
| 13 | GISS-E2-H | (Schmidt et al., 2006) |
| 14 | GISS-E2-R | (Schmidt et al., 2006) |
| 15 | HadGEM2-ES | (Collins et al., 2011) |
| 16 | IPSL-CM5A-LR | (Dufresne et al., 2013) |
| 17 | IPSL-CM5A-MR | (Dufresne et al., 2013) |
| 18 | IPSL-CM5B-LR | (Dufresne et al., 2013) |
| 19 | MIROC-ESM | (Watanabe et al., 2011) |
| 20 | MIROC5 | (Watanabe et al., 2010) |
| 21 | MPI-ESM-LR | (Giorgetta et al., 2013) |
| 22 | MPI-ESM-MR | (Giorgetta et al., 2013) |
| 23 | MPI-ESM-P | (Giorgetta et al., 2013) |
| 24 | MRI-CGCM3 | (Yukimoto et al., 2012) |
| 25 | NorESM1-M | (Bentsen et al., 2013; Iversen et al., 2013) |
| 26 | bcc-csm1-1 | (Wu et al., 2014) |
| 27 | bcc-csm1-1-m | (Wu et al., 2014) |
| 28 | inmcm4 | (Volodin et al., 2010) |

**Table 2: List of CMIP5 models alongside the index used in the figures of this study and a reference.**



| Index used in plots | Model | Reference |
|---|---|---|
| 29 | AWI-CM-1-1-MR | (Rackow et al., 2018; Sidorenko et al., 2015) |
| 30 | BCC-CSM2-MR | (Wu et al., 2019) |
| 31 | BCC-ESM1 | (Wu et al., 2019) |
| 32 | CAMS-CSM1-0 | (Rong et al., 2018) |
| 33 | CESM2 | (Gettelman et al., 2019b) |
| 34 | CESM2-WACCM | (Gettelman et al., 2019b) |
| 35 | CNRM-CM6-1 | (Voldoire et al., 2019) |
| 36 | CNRM-CM6-1-HR | (Voldoire et al., 2019) |
| 37 | CNRM-ESM2-1 | (Séférian et al., 2019) |
| 38 | CanESM5 | (Swart et al., 2019) |
| 39 | E3SM-1-0 | (Golaz et al., 2019) |
| 40 | EC-Earth3-Veg | (Wyser et al., 2019) |
| 41 | FGOALS-f3-L | (Guo et al., 2020; He et al., 2019; He et al., 2020) |
| 42 | GFDL-CM4 | (Held et al., 2019) |
| 43 | GFDL-ESM4 | (Held et al., 2019) |
| 44 | GISS-E2-1-G | (Rind et al., 2020) |
| 45 | GISS-E2-1-H | (Rind et al., 2020) |
| 46 | HadGEM3-GC31-LL | (Kuhlbrodt et al., 2018) |
| 47 | INM-CM4-8 | (Volodin et al., 2017a; Volodin et al., 2017b) |
| 48 | INM-CM5-0 | (Volodin et al., 2017a; Volodin et al., 2017b) |
| 49 | IPSL-CM6A-LR | (Boucher et al.) |
| 50 | MCM-UA-1-0 | (Delworth et al., 2002) |
| 51 | MIROC-ES2L | (Hajima et al., 2020) |
| 52 | MIROC6 | (Tatebe et al., 2019) |
| 53 | MPI-ESM1-2-HR | (Muller et al., 2018) |
| 54 | MRI-ESM2-0 | (Yukimoto et al., 2019) |
| 55 | NESM3 | (Cao et al., 2018) |
| 56 | NorESM2-LM | (Seland et al., 2020) |
| 57 | SAM0-UNICON | (Park et al., 2019) |
| 58 | UKESM1-0-LL | (Sellar et al., 2019) |

**Table 3: As Table 2 but for CMIP6 models included in this study.**





| Label | ECS (original publication) | ECS (CMIP5) [K] | ECS (CMIP6) [K] | $p$ (CMIP5) | $p$ (CMIP6) |
|---|---|---|---|---|---|
| **BRI** | most likely 4.0 K, < 2.30 K very unlikely (90% confidence) | 3.72 ± 0.56 | 4.36 ± 1.16 | 0.0002 | 0.0219 |
| **COX** | 2.8 K ± 0.6 K | 3.03 ± 0.71 | 3.44 ± 1.15 | 0.0010 | 0.0545 |
| **LIP** | no best estimate given | 2.97 ± 0.76 | 3.66 ± 1.27 | 0.0043 | 0.2039 |
| **SHD** | none – see SHL | 3.65 ± 0.63 | 3.74 ± 1.11 | 0.0006 | 0.1120 |
| **SHL** | most likely 4 K with lower limit 3 K | 3.42 ± 0.63 | 3.65 ± 1.05 | 0.0001 | 0.0114 |
| **SHS** | none – see SHL | 3.07 ± 0.68 | 3.41 ± 1.13 | 0.0581 | 0.0638 |
| **SU** | most likely 4 K with lower limit 3 K | 3.30 ± 0.90 | 3.69 ± 1.59 | 0.0919 | 0.8573 |
| **TIH** | most likely 4.0 K | 3.88 ± 0.78 | 4.07 ± 1.21 | 0.0002 | 0.0454 |
| **TII** | most likely 4.0 K | 3.87 ± 0.70 | 4.11 ± 1.17 | 0.0004 | 0.0634 |
| **VOL** | 3.6 K ± 0.4 K (standard deviation) | 3.74 ± 0.64 | 4.14 ± 1.13 | 0.0004 | 0.0057 |
| **ZHA** | 3.90 K ± 0.45 K (standard deviation) | 3.35 ± 0.72 | 3.81 ± 0.60 | 0.1195 | 0.0022 |

**Table 4: Overview of the constrained ECS ranges and p-values for all 11 analyzed emergent constraints. If not further specified, the uncertainty ranges correspond to the IPCC's "likely range" (66% confidence). For CMIP5 and CMIP6, these are evaluated from the probability distribution given by equation (8) (see also middle columns of Figure 2 to Figure 5). Note that even though CMIP5 models were used for some constraints in the original publications, the constrained ranges in column 2 and column 3 might differ due to the use of different models (in this paper, we use output from all CMIP models that is publicly available, see data availability section for details). The p-values describing the significance of the emergent relationships are defined as the probability that $r$ (given by the bootstrapped distribution) exhibits the opposite sign as originally expected from the emergent relationship (see shaded areas in the right columns of Figure 2 to Figure 5). Smaller p-values point to higher significance and vice versa (for details see section 2.3).**



## 7 Figures

Figure 1: Assessed ECS ranges (blue bars) from the Charney report (Charney et al., 1979) and the different Assessment Reports (ARs) of the Intergovernmental Panel on Climate Change (IPCC). The numbers correspond to individual CMIP5 and CMIP6 models; see Table A1 and Table A2 for details. Adapted and updated from Meehl et al. (2020).





**Figure 2: Emergent constraints BRI, COX and LIP applied to the CMIP5 ensemble (blue) and CMIP6 ensemble (orange). Left column: emergent relationships (solid blue and orange lines) for the CMIP models (numbers for individual models are specified in Table A1 and Table A2). The shaded areas around the regression lines correspond to the standard prediction errors (equation (5)), which defines the error in the regression model itself. The vertical black dashed line corresponds to the observational reference (see Table 1 for details on the individual observational datasets used) with its uncertainty range given as standard error (gray shaded area). The horizontal dashed lines show the best estimates of the constrained ECS for CMIP5 (blue) and CMIP6 (orange). The colored dots mark the CMIP5 (blue) and CMIP6 (orange) multi-model means. Middle column: probability densities for the constrained ECS following equation (8) (solid lines) and the unconstrained model ensembles (histograms). Note that for each individual emergent constraint, a different subset of climate is used due to the availability of data (see Table A1 and Table A2 for details). Thus, these histograms may differ for the different constraints. Right column: probability distributions of the linear (Pearson) correlation coefficient $r$ for 100,000 bootstrap samples (solid lines). The vertical solid lines correspond to the value of $r$ when the complete original sample is used for its calculation. The vertical gray line illustrates $r = 0$, i.e. no correlation between the emergent constraint x-axis and ECS. The shaded areas correspond to the $p$-values describing the significance of the emergent relationships. These are defined as the probability that $r$ (given by the bootstrapped distribution) exhibits the opposite sign as originally expected from the emergent relationship (see section 2.3). Smaller $p$-values point to higher significance and vice versa.**





**Figure 3: As Figure 2, but for the constraints SHD, SHL and SHS.**





**Figure 4: As Figure 2, but for the constraints SU, TIH and TII.**





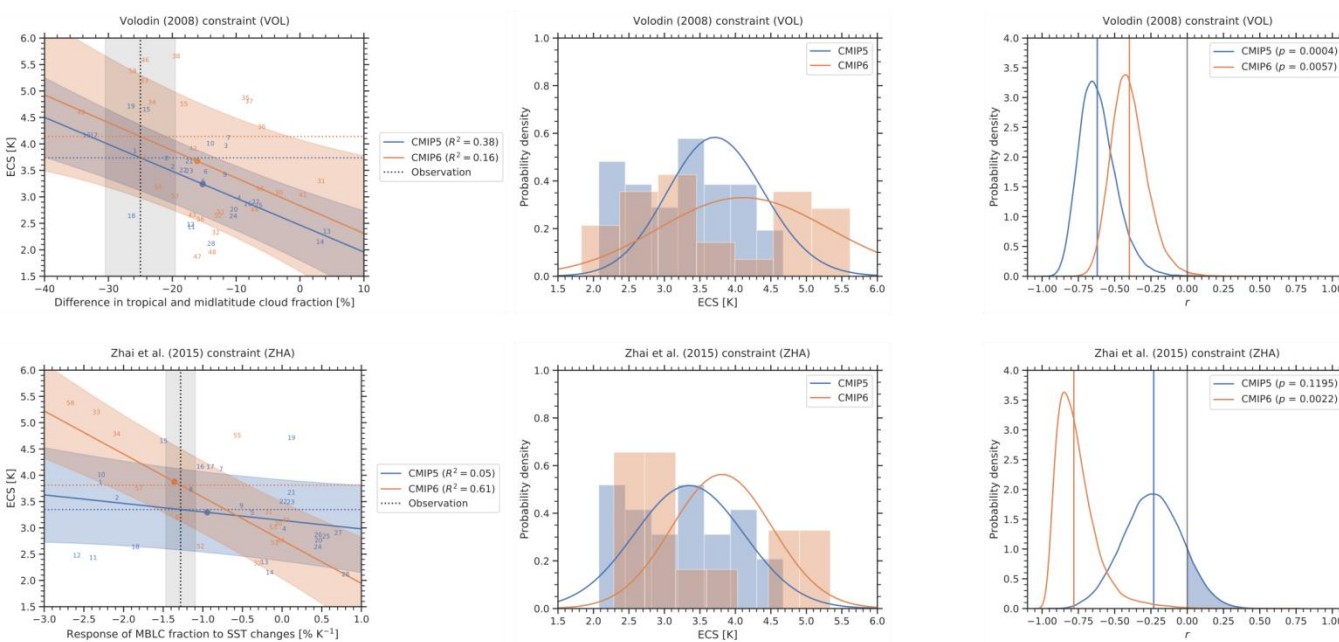

**Figure 5: As Figure 2, but for the constraints VOL and ZHA.**




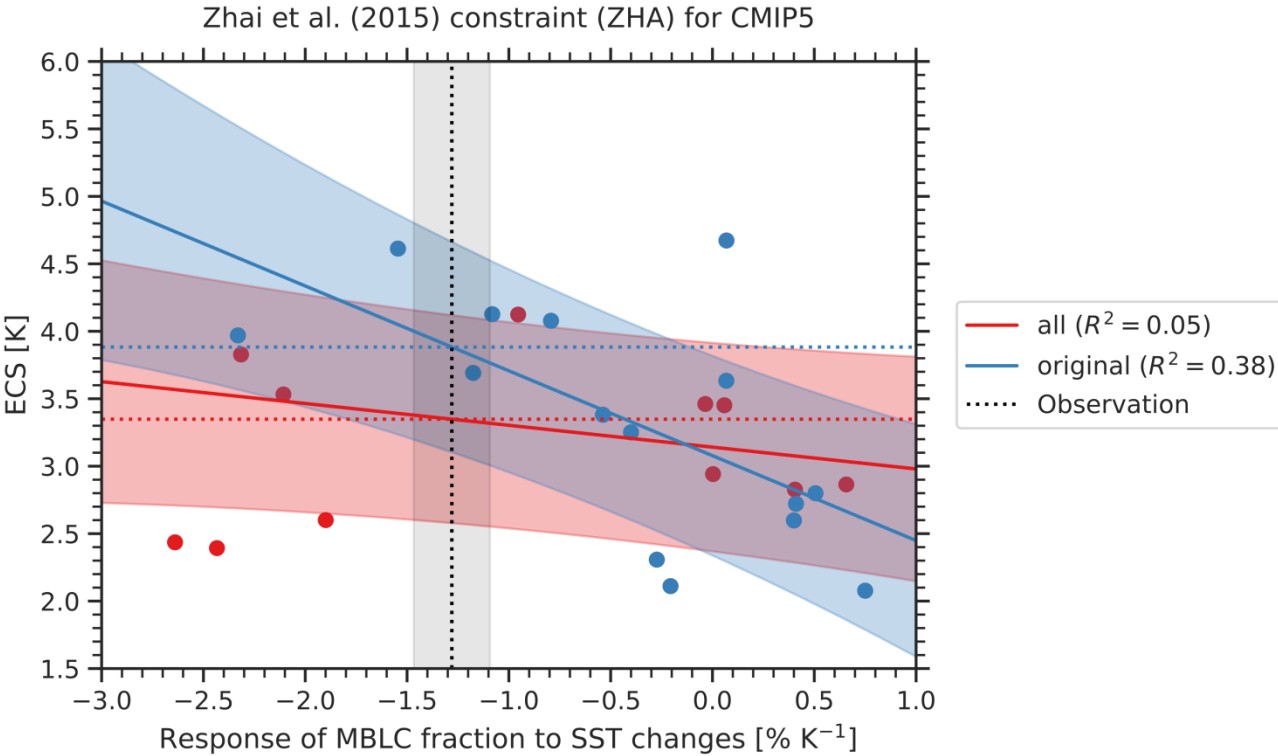

**Figure 6: Emergent relationship ZHA (Zhai et al., 2015) for different subsets of CMIP5 models. Blue circles show the 15 CMIP5 models used in the original publication (except for CESM1-CAM5): the solid blue line and blue shaded area show the emergent relationships evaluated on these models including the uncertainty range. In our study, we added 11 more CMIP5 models (red circles). The corresponding emergent relationship that considers all available CMIP5 models is shown in red colors. This**
**relationship shows a considerable lower coefficient of determination ($R^2$) than the relationship using the original subset of CMIP5 models. The vertical dashed line and shaded area correspond to the observational reference and the horizontal dashed lines to the corresponding ECS constraints using this observation.**




**Figure 7: Boxplots of the distribution of the 100,000 bootstrap samples of the linear (Pearson) correlation coefficient $r$ for the different emergent constraints (x-axis) evaluated on different climate model ensembles (different colors). For better transparency, the sign of correlation coefficients from emergent relationships with expected negative slopes have been changed. Thus, positive values of $r$ indicate that $r$ matches the sign of the expected relationship, while negative values indicate that $r$ does not match the expected sign. The "x" shows the mean of the distribution, the horizontal central line the median, the edges of the box the 25% and 75% percentile and the whiskers the 5% and 100% percentiles which form the one-sided 95% confidence interval of $r$. An emergent relationship is highly or barely significant on a climate model ensemble if the one-sided 95% confidence interval does not include $r = 0$ (horizontal gray line). Otherwise it is called "almost significant" or "far from significant" (see section 2.3 for details). Only four emergent constraints are either highly or barely significant on both climate model ensembles: BRI, SHL, TIH and VOL.**



**8   Appendix A**

|    | Model       | ECS  | BRI   | COX  | LIP    | SHD  | SHL  | SHS  | SU   | TIH   | TII   | VOL    | ZHA   |
|----|-------------|------|-------|------|--------|------|------|------|------|-------|-------|--------|-------|
| 1  | ACCESS1-0   | 3.83 | -1.59 | 0.20 | -33.70 | 0.45 | 0.84 | 0.39 | 0.94 | 14.14 | 0.54  | -26.16 | -2.32 |
| 2  | ACCESS1-3   | 3.53 | -1.59 | 0.15 | -34.32 | 0.54 | 0.90 | 0.36 | 0.93 | 21.05 | 0.64  | -20.21 | -2.11 |
| 3  | BNU-ESM     | 3.92 | -1.39 | 0.14 | -33.38 | 0.35 | 0.75 | 0.41 | 1.11 | 22.56 | 1.47  | -11.91 |       |
| 4  | CCSM4       | 2.94 | -0.09 | 0.18 | -37.64 | 0.39 | 0.75 | 0.37 |      | 24.32 | 1.17  | -9.88  | 0.00  |
| 5  | CNRM-CM5    | 3.25 | 0.29  | 0.15 | -34.53 | 0.38 | 0.71 | 0.33 | 1.04 | 14.54 | 1.50  | -15.43 | -0.40 |
| 6  | CNRM-CM5-2  | 3.44 | 0.56  |      | -34.62 | 0.40 | 0.73 | 0.33 |      | 15.83 | 1.44  | -15.08 |       |
| 7  | CSIRO-Mk3-6-0 | 4.08 | -1.57 | 0.20 | -35.12 | 0.61 | 0.97 | 0.36 | 1.00 | 11.19 | 1.01 | -11.53 | -0.79 |
| 8  | CanESM2     | 3.69 | -0.72 | 0.16 | -34.28 | 0.30 | 0.61 | 0.31 | 1.00 | 20.00 | 0.69  | -21.21 | -1.18 |
| 9  | FGOALS-g2   | 3.38 | 0.33  | 0.05 | -30.75 | 0.29 | 0.79 | 0.51 | 1.26 | 24.61 | 0.93  | -12.06 | -0.54 |
| 10 | GFDL-CM3    | 3.97 | -0.26 | 0.32 | -35.03 | 0.33 | 0.67 | 0.34 | 1.01 | 18.05 | 1.43  | -14.67 | -2.33 |
| 11 | GFDL-ESM2G  | 2.39 | -0.80 | 0.19 | -36.47 | 0.30 | 0.76 | 0.45 | 0.93 | 18.60 | 0.74  | -17.63 | -2.43 |
| 12 | GFDL-ESM2M  | 2.44 | -0.34 | 0.15 | -35.95 | 0.25 | 0.68 | 0.43 | 0.94 | 17.43 | 1.36  | -17.75 | -2.64 |
| 13 | GISS-E2-H   | 2.31 | 1.18  | 0.10 | -33.87 | 0.32 | 0.61 | 0.29 | 0.71 | 21.70 | 2.58  | 3.57   | -0.27 |
| 14 | GISS-E2-R   | 2.11 | 0.89  | 0.11 | -34.14 | 0.32 | 0.60 | 0.28 | 0.71 | 12.88 | 2.13  | 2.52   | -0.21 |
| 15 | HadGEM2-ES  | 4.61 | -2.45 | 0.26 | -34.58 | 0.43 | 0.81 | 0.38 | 0.95 | 11.54 | 0.87  | -24.69 | -1.54 |
| 16 | IPSL-CM5A-LR | 4.13 | -0.96 | 0.21 | -32.13 | 0.41 | 0.86 | 0.45 | 0.95 | 11.58 | 0.53 | -34.04 | -1.08 |
| 17 | IPSL-CM5A-MR | 4.12 | -1.18 | 0.15 | -33.61 | 0.48 | 0.92 | 0.44 | 0.98 | 5.73  | 0.36 | -32.90 | -0.95 |
| 18 | IPSL-CM5B-LR | 2.60 | -0.50 | 0.16 | -32.02 | 0.25 | 0.67 | 0.41 | 0.89 | 21.25 | 0.92 | -27.02 | -1.90 |
| 19 | MIROC-ESM   | 4.67 | -0.99 | 0.22 | -31.87 | 0.33 | 0.89 | 0.56 | 0.95 | 0.94  | -0.54 | -27.09 | 0.07  |
| 20 | MIROC5      | 2.72 | 0.27  | 0.22 | -35.51 | 0.36 | 0.78 | 0.42 | 1.01 | 4.90  | 0.40  | -10.99 | 0.41  |
| 21 | MPI-ESM-LR  | 3.63 | -0.35 | 0.15 | -34.67 | 0.41 | 0.86 | 0.45 | 1.07 | 3.16  | 0.24  | -17.96 | 0.07  |
| 22 | MPI-ESM-MR  | 3.46 | -0.49 | 0.16 | -34.33 | 0.42 | 0.87 | 0.45 | 1.04 | 6.10  | 0.30  | -18.92 | -0.04 |
| 23 | MPI-ESM-P   | 3.45 | -0.68 |      | -34.36 | 0.41 | 0.87 | 0.46 |      | -0.23 | 0.13  | -17.97 | 0.06  |
| 24 | MRI-CGCM3   | 2.60 | -1.08 | 0.09 | -35.01 | 0.37 | 0.78 | 0.41 | 1.04 | 34.02 | 2.04  | -11.08 | 0.40  |
| 25 | NorESM1-M   | 2.80 | -0.74 | 0.13 | -37.44 | 0.44 | 0.82 | 0.38 | 1.07 | 26.23 | 0.65  | -7.13  | 0.51  |
| 26 | bcc-csm1-1  | 2.83 | -0.11 | 0.18 | -34.25 | 0.37 | 0.78 | 0.41 | 1.15 | 30.84 | 1.34  | -8.77  | 0.41  |
| 27 | bcc-csm1-1-m | 2.86 | -0.46 | 0.13 | -36.36 | 0.34 | 0.74 | 0.40 | 1.19 | 43.64 | 2.69 | -7.56  | 0.66  |
| 28 | inmcm4      | 2.08 | -0.18 | 0.07 | -36.43 | 0.19 | 0.52 | 0.33 | 0.84 | 28.16 | 1.89  | -14.52 | 0.75  |

**Table A1: All participating CMIP5 models including their ECS values (in K) and x-axis values for the different emergent constraints. Details on all constraints (including the units) are given in Table 1. The leftmost column corresponds to the index used in all plots.**





| | Model | ECS | BRI | COX | LIP | SHD | SHL | SHS | SU | TIH | TII | VOL | ZHA |
|---|---|---|---|---|---|---|---|---|---|---|---|---|---|
| 29 | AWI-CM-1-1-MR | 3.16 | | 0.12 | -35.43 | 0.42 | 0.76 | 0.34 | 1.05 | 2.15 | | | |
| 30 | BCC-CSM2-MR | 3.04 | -1.43 | 0.16 | -36.12 | 0.54 | 0.89 | 0.36 | 1.10 | 32.92 | 1.50 | -3.93 | -0.10 |
| 31 | BCC-ESM1 | 3.26 | | 0.19 | -36.18 | 0.46 | 0.88 | 0.42 | 1.18 | 38.93 | 0.94 | 2.69 | -0.22 |
| 32 | CAMS-CSM1-0 | 2.29 | -0.44 | 0.14 | -35.38 | 0.47 | 0.91 | 0.44 | 1.09 | 27.52 | 1.40 | -13.84 | -0.36 |
| 33 | CESM2 | 5.16 | -1.94 | 0.15 | -36.22 | 0.49 | 0.85 | 0.36 | 0.93 | -7.25 | 0.45 | -24.99 | -2.39 |
| 34 | CESM2-WACCM | 4.75 | -1.60 | 0.18 | -36.43 | 0.55 | 0.91 | 0.36 | 0.93 | -5.37 | -0.13 | -23.81 | -2.14 |
| 35 | CNRM-CM6-1 | 4.83 | -0.61 | 0.11 | -34.50 | 0.30 | 0.74 | 0.44 | 1.03 | 16.59 | 0.92 | -9.16 | |
| 36 | CNRM-CM6-1-HR | 4.28 | -0.25 | 0.11 | -33.41 | 0.34 | 0.81 | 0.46 | 1.04 | 13.31 | 1.04 | -6.63 | |
| 37 | CNRM-ESM2-1 | 4.76 | -0.51 | 0.08 | -34.83 | 0.30 | 0.74 | 0.44 | 1.04 | 16.17 | 0.95 | -8.60 | |
| 38 | CanESM5 | 5.62 | -0.97 | 0.18 | -35.50 | 0.48 | 0.82 | 0.34 | 0.97 | 15.39 | 0.04 | -20.02 | |
| 39 | E3SM-1-0 | 5.32 | | 0.22 | -35.59 | | | | | 16.73 | 1.26 | | |
| 40 | EC-Earth3-Veg | 4.31 | | 0.19 | | | | | | | 0.82 | | |
| 41 | FGOALS-f3-L | 3.00 | | 0.09 | | 0.31 | 0.76 | 0.44 | 1.06 | 11.75 | 1.00 | -0.19 | |
| 42 | GFDL-CM4 | 3.87 | -1.06 | 0.16 | -34.14 | 0.49 | 0.92 | 0.42 | 1.03 | 3.08 | 0.32 | -17.36 | |
| 43 | GFDL-ESM4 | 2.62 | | 0.15 | -34.82 | 0.48 | | | | 6.07 | 0.64 | -17.53 | |
| 44 | GISS-E2-1-G | 2.72 | -0.58 | 0.17 | -36.24 | 0.26 | 0.61 | 0.35 | 0.98 | 28.46 | 2.39 | -7.75 | -0.07 |
| 45 | GISS-E2-1-H | 3.11 | -0.48 | 0.13 | -36.95 | 0.24 | 0.59 | 0.36 | 1.00 | 24.55 | 1.84 | -6.82 | 0.00 |
| 46 | HadGEM3-GC31-LL | 5.55 | -0.56 | 0.20 | -35.75 | 0.46 | 0.92 | 0.46 | 1.02 | 9.56 | 0.99 | -24.87 | |
| 47 | INM-CM4-8 | 1.83 | 0.03 | 0.13 | -38.15 | 0.47 | 0.71 | 0.25 | 0.99 | 12.22 | 0.62 | -16.71 | |
| 48 | INM-CM5-0 | 1.92 | -0.46 | 0.14 | -37.34 | 0.40 | 0.63 | 0.23 | 0.99 | 11.22 | 0.62 | -14.38 | |
| 49 | IPSL-CM6A-LR | 4.56 | -0.38 | 0.14 | -33.74 | 0.50 | 0.92 | 0.41 | 0.76 | 11.90 | 0.41 | -34.90 | |
| 50 | MCM-UA-1-0 | 3.65 | | 0.20 | | | | | | | -0.56 | | |
| 51 | MIROC-ES2L | 2.68 | -0.49 | 0.15 | -34.13 | 0.36 | 0.79 | 0.43 | 0.91 | 18.53 | 0.68 | -13.03 | -0.14 |
| 52 | MIROC6 | 2.61 | -0.45 | 0.17 | -34.90 | 0.46 | 0.91 | 0.45 | 0.99 | 5.79 | 0.29 | -13.43 | -1.08 |
| 53 | MPI-ESM1-2-HR | 2.98 | -0.34 | 0.15 | -34.60 | 0.45 | 0.78 | 0.33 | 1.04 | 4.33 | 0.53 | -20.21 | -0.16 |
| 54 | MRI-ESM2-0 | 3.15 | -0.73 | 0.15 | -33.39 | 0.39 | 0.79 | 0.40 | 0.95 | 3.33 | 0.75 | -22.83 | -1.35 |
| 55 | NESM3 | 4.72 | -0.73 | 0.24 | -36.77 | 0.52 | 0.89 | 0.37 | 1.05 | 28.78 | 0.55 | -18.81 | -0.62 |
| 56 | NorESM2-LM | 2.54 | | 0.20 | | | | | 1.01 | 34.40 | 1.30 | -16.22 | |
| 57 | SAM0-UNICON | 3.72 | -1.40 | 0.19 | -36.41 | 0.54 | 0.90 | 0.36 | 1.01 | 11.99 | 1.13 | -25.25 | -1.85 |
| 58 | UKESM1-0-LL | 5.34 | -0.59 | 0.21 | -35.67 | 0.48 | 0.93 | 0.45 | 1.03 | 2.22 | 0.86 | -26.83 | -2.72 |

**Table A2: As Table A1, but for the CMIP6 models.**





## 9    Code availability

The corresponding ESMValTool recipe that can be used to reproduce the figures of this paper will be included in ESMValTool v2.0 (Eyring et al., 2020; Lauer et al., 2020; Righi et al., 2020) at the time of publication of this paper. ESMValTool v2.0 is released under the Apache License, VERSION 2.0. The latest release of ESMValTool v2.0 is publicly available on Zenodo at https://doi.org/10.5281/zenodo.3401363. The source code of the ESMValCore package, which is

installed as a dependency of the ESMValTool v2.0, is also publicly available on Zenodo at https://doi.org/10.5281/zenodo.3387139. ESMValTool and ESMValCore are developed on the GitHub repositories available at https://github.com/ESMValGroup.

## 10    Data availability

CMIP5 and CMIP6 model output (see Table 2 and Table 3) is available through the Earth System Grid Foundation (ESGF)

and can be directly used within the ESMValTool (e.g. https://esgf-data.dkrz.de/projects/esgf-dkrz/). Downloading instructions and preprocessing scripts for the observational datasets detailed in Table 1 are included in the ESMValTool distribution.

## 11    Author contribution

MS led the writing and analysis of the paper. MS and AL coded the emergent constraints in the ESMValTool. VE, PG and

SCS contributed to the concept of the study and the interpretation of the results. All authors contributed to the writing of the manuscript.

## 12    Competing interests

The authors declare that they have no conflict of interest.

## 13    Acknowledgements

This work has been supported by the European Union's Horizon 2020 Framework Programme for Research and Innovation "Coordinated Research in Earth Systems and Climate: Experiments, kNowledge, Dissemination and Outreach (CRESCENDO)" project under Grant Agreement No. 641816, the European Union's Horizon 2020 project "Climate-Carbon Interactions in the Coming Century" (4C) under Grant Agreement No. 821003 and the "Advanced Earth System Model Evaluation for CMIP (EVal4CMIP)" project funded by the Helmholtz Society. We acknowledge the World Climate

Research Programme (WCRP), which, through its Working Group on Coupled Modelling, coordinated and promoted





CMIP5 and CMIP6. We thank the climate modeling groups for producing and making available their model output, the Earth System Grid Federation (ESGF) for archiving the data and providing access, and the multiple funding agencies who support CMIP and ESGF. The computational resources of the Deutsches Klimarechenzentrum (DKRZ, Hamburg, Germany) that allowed the analysis of this study are kindly acknowledged.

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
