# Peer review of "Emergent constraints on Equilibrium Climate Sensitivity in CMIP5: do they hold for CMIP6?"

_Earth System Dynamics, 2020_

## Referee Comment (RC1) · Peter Caldwell (Referee) · 19 Aug 2020

This study confronts 11 emergent constraints on ECS with CMIP6 data for the first time. The skill of most of these constraints collapses when faced with new data. All constraints predict higher ECS based on CMIP6 data relative to CMIP5 because many CMIP6 models have higher ECS yet similar constraint values. Overall I thought this paper was excellent, well-written, and very worthy of publication. I think the statistical significance methodology is inappropriate, however, which will require substantial revision. Otherwise I have a somewhat large number of fairly minor comments.

Major comments (in order of importance):

1. I'm uncomfortable with your bootstrap statistical significance testing method in sect

2.3.

1a. Your definition of statistical significance as "the sensitivity of the regression model to changes in the input data, i.e. the removal or addition of datasets" seems overly narrow. I think of statistical significance in this case as the probability of obtaining a correlation magnitude of at least r under the null hypothesis that no real correlation exists. By using such a narrow definition, I think you've avoided thinking about whether your methodology fully captures all sources of uncertainty. Your test is also weird in that it lacks any sense of a null hypothesis that there is no real correlation. Instead, you seem to just be taking the correlation obtained with all models and creating a histogram of possible values for it by recomputing correlations with models added or removed. I don't think this is appropriate.

1b. I suspect your bootstrapping results are strongly dependent on arbitrary sampling design choices: removing a model or adding multiple copies of a model makes a huge difference to your regression when you only have ∼30 models in the CMIP archive with data for a particular constraint. Thus I expect the number of random samples you draw to have a big impact on your histogram of bootstrapped correlations. To disprove my complaint, you could create histograms with M-2, M-1, M, M+1, and M+2 models. If these all look the same, then my criticism is misplaced.

1c. It seems more defensible - or at least complimentary - to use a T-test as described in https://atmos.uw.edu/∼dennis/552_Notes_3.pdf. If you did use the T-test, would you get similar results?

2. You don't mention any of the limitations of your methodology until the conclusions section. This left me reading through the methodology and results sections under the impression that you were unaware of the possible flaws with what you were doing. It would be helpful for readers if you describe potential problems with the methodology in the methodology section so readers won't traverse the paper thinking you don't know what you're doing and so they can interpret your results with an appropriate level of

skepticism.

2a.  In addition to the problems with your methodology you currently mention in the conclusions, it is probably worth also mentioning that giving the models which agree worst with the observed constraint value equal weight in determining the regression is probably a bad idea.  This issue is explained nicely in Brient (2020; https://link.springer.com/article/10.1007%2Fs00376-019-9140-8)

3. Caldwell et al (2018; https://journals.ametsoc.org/jcli/article/31/10/3921/94898/Evaluating-Emergent-Constraints-on-Equilibrium) tested 5 constraints trained on earlier CMIP ensembles on CMIP5 data and found that 4 of these constraints (Covey, Trenberth, and Fasullo D and M) also failed when confronted with out-of-sample data.  In that context, I see your paper as a follow-up to the Caldwell paper. I think it would be worth mentioning this around L60.  It is interesting that Volodin was trained on CMIP3 data but also holds up for CMIP5 and CMIP6 data.

4. On a related note, I felt you undersold the importance of Zhai failing when confronted with CMIP5 data which it wasn't originally trained on.  A similar thing happened in Caldwell et al 2018 with the Qu constraint. Such sensitivity to sampling details seems to me an important indicator that the number of models we have in the CMIP archive is insufficient for making robust conclusions about the credibility - or lack of credibility - of the constraints we propose.

5.       Your introduction argument that ECS hasn't changed in 40 yrs feels dated in light of Sherwood et al (2020; https://agupubs.onlinelibrary.wiley.com/doi/abs/10.1029/2019RG000678).     I know you didn't mention this study because it wasn't accepted when you submitted the paper, but should be cited in the revision.

6. L116: I'm pretty sure P(y|x)*P(x) can be written as a Gaussian function and therefore evaluated analytically rather than numerically integrated.  You might have to use the fact that $e^{x + C} = e^x * e^C$ for some constant C in conjunction with completing the

squares to manage this. This comment isn't a big deal - numerical integration is fine - but analytic integration is more elegant.

7. L118: I don't understand why you need to assume P(y|y0)=P(y0|y) in eq 6 and therefore that the prior is uniform. Perhaps you could explain this in more detail. As I see it, you are just assuming y has a Gaussian distribution with mean \hat{y} and variance sigma_{x_0}. These are definitely big assumptions, but don't imply a prior.

8. I got a bit lost regarding which of your results depend on the Gaussian approach of sect 2.2 for what results use the bootstrapping of sect 2.3. Am I correct that the left panels of Fig 2-5 use linear regression and the standard error, the middle panels of these same figures use the Gaussian approach and everything else is based on bootstrapping? It would be useful to mention at the end of sect 2.2 and 2.3 what figures use the methodology just described.

9. L156: how did you choose the 11 constraints you evaluate? Readers may think you cherry picked the constraints that behaved poorly if you don't say explicitly why you chose the ones you did.

10. L276 says Volodin was the first emergent constraint on ECS, which isn't true. Covey et al 2000 and Knutti et al 2006 provide earlier emergent constraints.

11. L433: Bretherton and Caldwell (2020; https://journals.ametsoc.org/jcli/article/33/17/7413/348548/Combining-Emergent-Constraints-for-Climate) provide a multivariate technique for combining constraints on ECS. Doing so provided less conceptual insight than I expected - having most constraints predict high ECS led to the combined estimate also having high ECS with narrower uncertainty... which seems obvious in retrospect.

Minor Comments:

1. L18 "which stem the major source" is wrong. I think you mean "which is the major source"?

2. You often say things like "the emergent-constrained best estimate". "Emergentconstrained" doesn't make sense. I think you mean the "emergent-constraint-constrained".

3. L66: you already gave the range of ECS in the previous line, so saying CMIP6 models exceed 5K is redundant/unnecessary.

4. Eq 3: x should either include or exclude "_m" on *both* sides of the equation.

5. Eq 8 uses P(y|x) from eq 6, which says it is an equation for P(y|x_0). I think eq 6 is really true for all x rather than just the observed value x_0. I suggest you remove mention of x_0 everywhere before eq 6.

6. Sect 2.3: does it really take 100,000 samples to characterize uncertainty in a correlation between the 20-50 samples you're getting from the CMIP archive? I would guess 1000 iterations would be sufficient.

7. L167: "Temperature (ERSST) is used": I'm confused because I thought you said you used HadISST on L164. Are you saying that the Brient + Schneider used ERSST?

8. L351: I've never seen the "(here: ...)" nomenclature you use. Do you mean "(e.g. ...)"?

---

## Referee Comment (RC2) · Thorsten Mauritsen (Referee) · 25 Aug 2020

Review of "Emergent constraints on Equilibrium climate sensitivity in CMIP5: do they hold for CMIP6?" by Manuel Schlund and co-authors.

In this study a series of mostly process-oriented emergent constraints that were developed on earlier model ensembles are applied to the latest CMIP6 ensemble. This is a very welcome attempt and in a broad sense testing scientific reproducibility. My major concern is with the main conclusions drawn, or perhaps not drawn, from the results. The fact that estimated ECS based on these constraints increases roughly in proportion to the mean ECS increase from CMIP5 to CMIP6 suggests that these constraints are in not actually constraints on ECS, rather, at best they are constraints on the feed-

back processes they target. I develop argumentation this below, along with providing some more technical comments. I sign this review such that should the authors have any issues understanding my point they can contact me directly.

Sincerely,

Thorsten Mauritsen

—

Climate sensitivity is inversely proportional to the feedback parameter (lambda in equation 2) which in turn is a sum of a series of processes (sum of lambda_i). My take on the situation is that many of the emergent constraints (except COX) were successful on CMIP5 because inter-model spread in ECS in that ensemble was dominated by spread in low-level cloud feedbacks in the tropics. However, if any other feedback, e.g. water vapour feedback or any other cloud feedback, is biased in the ensemble as a whole then these kinds of process-oriented emergent constraints will necessarily be biased in their estimates of ECS. Probably even collectively since they thrive on the same kind of model spread, and so just because there are many studies that agree doesn't increase our confidence in their quantitative outcome. Likewise, if structural commonalities among models cause an unreasonable low inter model spread in some other feedback process then the emergent constraint is going to be over-confident. All in all, the results suggest that the original studies were overly confident and that changes in feedbacks not constrained by these studies cause them to be biased with a sign that cannot be determined (since CMIP6 probably also contains collectively biased feedback processes). Thus, these process-oriented emergent constraints are perhaps best thought of as constraints on the processes that they target, rather than constraints on ECS, and in extension the original studies have been disproven by the results of this study.

There are alternatives to process-oriented emergent constraints, though, one of them which is included in this study (COX, more about this study and why I think there

is a shift below). Emergent constraints that use global temperature change as a predictor of ECS do not suffer from the same problem: even if one feedback is biased in a model ensemble the constraint can in principle still work since both global change and ECS are inversely proportional to the sum of feedbacks. Suggestions of emergent constraints of this kind include Last Glacial Maximum (Hargreaves et al. 2012 doi:10.1029/2012GL053872), Pliocene warming (Hargreaves and Annan, 2016 doi:10.5194/cp-12-1591-2016), and post-1970s warming (Jimenez-de-la-Cuesta and Mauritsen 2019 doi:10.1038/s41561-019-0463-y). All of these ideas have been tested across ensembles including CMIP6/PMIP4 (Tokarska et al. 2020 doi:10.1126/sciadv.aaz9549; Renoult et al. 2020 doi:10.5194/cp-2019-162), finding essentially unchanged results between ensembles. Other studies worth mentioning are Bender et al. (2010, doi:10.1007/s00382-010-0777-3) and Dessler and Forster (2018), although I haven't seen tests of these.

I think all of the above is rather straightforward and fairly easy to understand. I think the authors have everything at hand that they need to draw the conclusion that the process-oriented emergent constraints are not useful for estimating ECS, but rather should be better thought of as ideally constraining part of the cloud feedback. There are several places throughout that needs revising.

Other major points

I think it is not reasonable to provide best estimates of ECS in the abstract and summary based on this study for the following reasons:

1) The above issue.

2) Because the study does not apply the latest observations to the constraints, rather opts for using the original observations. This is a perfectly fine choice given the scope of the paper, but it does mean the constraints are not up to date.

3) Because the study uses an implicit flat prior which in case of weak data automatically
leads to high-biased results.

I would instead suggest the authors cite percentage increases which is anyway all that is relevant here.

Regarding the Cox et al. 2018 constraint, section 3.2, this is built on the Hasselmann (1976) single heat capacity model. In this model there is a linear relationship between Psi and ECS, however, and despite what they claim if you add a deep ocean to the model you obtain a non-linear relationship wherein the relationship is weaker for higher ECS, see Annan et al. (2020 doi:10.5194/esd-11-709-2020), their figure 5 (note flipped axes). If you look at how the CMIP6 models are distributed they are simply situated in the flatter part of the expected curve, and if you fit a straight line to it you will obtain different slopes than for CMIP5.

In this regard, and this applies not only to this study but most of these kinds, I am concerned with the general use of linear regression. The most silly example is SU constraint, where despite getting the wrong sign of the slope in CMIP6, you obtain a constraint on ECS. I think studies must be much more smart about their choice of statistical model, and not just use linear regression when non-linear behaviour is expected or other physical constraints can be applied such as a near-zero intercept, examples in Jimenez-de-la-cuesta-Otero and Mauritsen (2019), Annan et al. (2020) and Renoult et al. (2020). In case of process-oriented emergent constraints one could perhaps think of using Equation (2) in the form ECS $\sim$ a/(b+x) where x is a process-oriented predictor. I am not saying the authors need to change this, but it would be worthwhile acknowledging that using linear regression, heedlessly, can lead to misleading and over-confident results.

I found the discussion of statistical significance somewhat disturbing. The chosen thresholds seem purely subjective, as far as I can tell. I would suggest to delete this whole discussion which seem rather pretentious.

I found Sections 4 and 5 rather long and repetitive. I would suggest revising and sharpening.

Minor things

19, 'of spread in ECS among models'

36, 'concentration over pre-industrial levels'

69, I am not sure Forster et al. 2020 is correctly cited here

85-92, perhaps drop Delta from F, and when using a specific forcing in equation 2 write $F_{4x}$ or something?

90-93. did you account for model energy leakage and drift? Concerning drift, some do account for this, but is not always obvious if there is a best way, nevertheless you must document what you did.

120-123, there are also an intermediate option, e.g. the Cauchy prior used in Annan and Hargreaves (2011, Climatic Change). Regarding the uniform prior, please specify which cut-off you use.

154-155, this is a misinterpretation, the IPCC 'likely' statements refer to 66-100 percent probability.

158-159, I felt this statement could be made more informative by explaining that it is the covariance of clouds with surface temperature anomalies.

180, 'results to choices made in the analysis'

236-263, these constraints seem to have some legacy with Fasullo and Trenberth (2012, Science), perhaps worth mentioning if the authors agree?

276-277, it is incorrect that Volodin (2008) was the first emergent constraint on ECS, there is Covey et al. 2000 and Knutti et al. 2006 before then.

320, I would suggest deleting 'describing the real world'

354-355, The idea and strength of an emergent constraint is that you use something you can observe to predict ECS. It really shouldn't matter if a process is slightly different in the warmer 2xCO2 world.

357-359, same applies here

360-361, this statement goes further than Zelinka et al. 2020. I would suggest replacing 'dominated by' -> 'to some extent associated with'

402-403, as per my above argumentation, I would be very careful with making this statement.

409, this might also have been shown by Klocke et al. (2011), check.

421-426. the very same paper also shows that the ECS estimated from 4xCO2 runs is higher than twice that in 2xCO2 runs, and that the bias of the same order of magnitude.

---

## Author Comment (AC1) · 26 Sep 2020

Reviewer comments are given in **bold**, our answers in red.

**This study confronts 11 emergent constraints on ECS with CMIP6 data for the first time. The skill of most of these constraints collapses when faced with new data. All constraints predict higher ECS based on CMIP6 data relative to CMIP5 because many CMIP6 models have higher ECS yet similar constraint values. Overall I thought this paper was excellent, well-written, and very worthy of publication. I think the statistical significance methodology is inappropriate,**

[Figure]

**however, which will require substantial revision. Otherwise I have a somewhat large number of fairly minor comments.**

We thank the reviewer for the helpful and constructive comments. We have now revised our manuscript in light of these and the other reviewer's comments we have received. A pointwise reply is given below.

**Major comments (in order of importance)**

**1. I'm uncomfortable with your bootstrap statistical significance testing method in sect 2.3.**

**1a. Your definition of statistical significance as "the sensitivity of the regression model to changes in the input data, i.e. the removal or addition of datasets" seems overly narrow. I think of statistical significance in this case as the probability of obtaining a correlation magnitude of at least r under the null hypothesis that no real correlation exists. By using such a narrow definition, I think you've avoided thinking about whether your methodology fully captures all sources of uncertainty. Your test is also weird in that it lacks any sense of a null hypothesis that there is no real correlation. Instead, you seem to just be taking the correlation obtained with all models and creating a histogram of possible values for it by recomputing correlations with models added or removed. I don't think this is appropriate.**

**1b. I suspect your bootstrapping results are strongly dependent on arbitrary sampling design choices: removing a model or adding multiple copies of a model makes a huge difference to your regression when you only have ∼30 models in the CMIP archive with data for a particular constraint. Thus I expect the number of random samples you draw to have a big impact on your histogram**

[Figure]

**of bootstrapped correlations. To disprove my complaint, you could create histograms with M-2, M-1, M, M+1, and M+2 models. If these all look the same, then my criticism is misplaced.**

**1c. It seems more defensible - or at least complimentary - to use a T-test as described in https://atmos.uw.edu/~dennis/552_Notes_3.pdf. If you did use the T-test, would you get similar results?**

Following this comment and the review comment by Thorsten Mauritsen (2nd referee) we decided to remove the bootstrap significance testing from the paper. We agree that our original definition and implementation of statistical significance was not optimal and therefore replaced it with the $t$-test on the correlation coefficient as proposed. The null hypothesis is that no correlation exists between the predictor and ECS. In the new revised version of the manuscript, we now give $p$-values of the emergent relationships that correspond to the probability that the absolute correlation is larger than $|r|$ even though the null hypothesis is true, i.e. the true underlying correlation is zero. In addition, we do not use the $p$-values anymore to specify *absolute* significance (our categories "highly significant", "barely significant", etc. were arguably arbitrary), but only use these $p$-values to specify *relative* significance, i.e. to indicate whether the statistical significance changes when moving from CMIP5 to CMIP6. Consistent with our original bootstrapping approach, the $t$-test also shows that except for the ZHA constraint, all emergent relationships show a higher significance for the CMIP5 ensemble than for the CMIP6 ensemble.

**2. You don't mention any of the limitations of your methodology until the conclusions section. This left me reading through the methodology and results sections under the impression that you were unaware of the possible flaws with what you were doing. It would be helpful for readers if you describe potential problems with the methodology in the methodology section so readers won't traverse the paper thinking you don't know what you're doing and so they can**

**interpret your results with an appropriate level of skepticism.**

We moved the discussion of the limitations of our methodology to the methods sections. In the conclusions sections we now only refer briefly to the limitations that are discussed in detail in the methods section:

"Our analysis makes a number of simplifying assumptions common to other studies, such as model independence, discussed in sections 2.1 and 2.2. These assumptions affect the significance of emergent relationships and the PDFs of ECS based on a constraint. However, they do not affect our main conclusions here, which concern the change in performance on CMIP6 relative to CMIP5 and the implications for robustness and future use of emergent constraints."

**2a. In addition to the problems with your methodology you currently mention in the conclusions, it is probably worth also mentioning that giving the models which agree worst with the observed constraint value equal weight in determining the regression is probably a bad idea. This issue is explained nicely in Brient (2020; https://link.springer.com/article/10.1007%2Fs00376-019-9140-8).**

We added the following discussion of this issue and the reference to the limitations paragraph at the end of the methods section:

"Moreover, our approach assigns equal model weights without taking model performance into account, i.e. agreement with observations. This issue is discussed in detail by Brient (2020)."

**3. Caldwell et al (2018; https://journals.ametsoc.org/jcli/article/31/10/3921/94898/ Evaluating-Emergent-Constraints-on-Equilibrium) tested 5 constraints trained on earlier CMIP ensembles on CMIP5 data and found that 4 of these constraints (Covey, Trenberth, and Fasullo D and M) also failed when confronted with out-of-sample data. In that context, I see your paper as a follow-up to the Caldwell**

**paper. I think it would be worth mentioning this around L60. It is interesting that Volodin was trained on CMIP3 data but also holds up for CMIP5 and CMIP6 data.**

We added your suggestion to the introduction:

"In addition, Caldwell et al. (2018) performed out-of-sample tests on five emergent constraints originally trained on older CMIP versions, by applying them to the CMIP5 ensemble. They found that out only one of the five passed this test. In this paper, we follow up on the work of Caldwell et al. (2018) by analyzing 11 published emergent constraints on ECS [...]".

**4. On a related note, I felt you undersold the importance of Zhai failing when confronted with CMIP5 data which it wasn't originally trained on. A similar thing happened in Caldwell et al 2018 with the Qu constraint. Such sensitivity to sampling details seems to me an important indicator that the number of models we have in the CMIP archive is insufficient for making robust conclusions about the credibility - or lack of credibility - of the constraints we propose.**

We added a short paragraph to section 5 (summary) that picks up on the failing ZHA constraint by referencing the corresponding Figure 6 and discussing that this result suggests that the credibility of the all other analyzed emergent constraints might be impaired:

"Moreover, our more detailed analysis of the ZHA constraint (see Figure 6) showed that this emergent constraint is very sensitive to outliers and the subset of the climate model ensemble used to fit the emergent relationship. Such a behavior might not be unique to the ZHA constraint but could apply to other emergent constraints as well. This in turn suggests that the number of climate models commonly used for emergent constraints might be too low leading to non-robust relationships."

**5. Your introduction argument that ECS hasn't changed in 40 yrs feels dated**

in light of Sherwood et al (2020; https://agupubs.onlinelibrary.wiley.com/doi/abs/ 10.1029/2019RG000678). I know you didn't mention this study because it wasn't accepted when you submitted the paper, but should be cited in the revision.

We added the following sentence to the introduction:

"A new assessment using this evidence has narrowed the 66% range (17–83%) to 2.6–3.9 K (Sherwood et al., 2020), but in the mean time CMIP6 models have a wider range (see below).

**6. L116: I'm pretty sure P(y|x)*P(x) can be written as a Gaussian function and therefore evaluated analytically rather than numerically integrated. You might have to use the fact that $e^{x+C} = e^x \cdot e^C$ for some constant C in conjunction with completing the squares to manage this. This comment isn't a big deal - numerical integration is fine - but analytic integration is more elegant.**

Here, $P(y|x) \cdot P(x)$ cannot be written as a simple Gaussian function in $x$ since $P(y|x)$ is not a Gaussian function in $x$ itself (only in $y$ when $x$ is held constant): the variance in $P(y|x)$ non-trivially depends on $x$ (equations (5) and (6)). Thus, the integration over $x$ of $P(y|x) \cdot P(x)$ cannot be done analytically (to the best of our knowledge) and must be done numerically.

**7. L118: I don't understand why you need to assume P(y|y0)=P(y0|y) in eq 6 and therefore that the prior is uniform. Perhaps you could explain this in more detail. As I see it, you are just assuming y has a Gaussian distribution with mean $\hat{y}$ and variance $\sigma_{x_0}$. These are definitely big assumptions, but don't imply a prior.**

We changed the manuscript accordingly:

"In this derivation of the probability $P(y)$ we do not assume any prior knowledge on ECS – in other words, that an ECS near 8 K would be deemed just as probable as one near 4 K if both are equally consistent with the observational best estimate $x_0$.

We do this for simplicity. The PDFs would shift somewhat lower with a broad prior on processes instead (see Sherwood et al. (2020)), but we are concerned here with how outcomes compare using CMIP5 vs. CMIP6 data, rather than the exact ranges obtained. Such comparisons are not sensitive to the prior."

**8. I got a bit lost regarding which of your results depend on the Gaussian approach of sect 2.2 for what results use the bootstrapping of sect 2.3. Am I correct that the left panels of Fig 2-5 use linear regression and the standard error, the middle panels of these same figures use the Gaussian approach and everything else is based on bootstrapping? It would be useful to mention at the end of sect 2.2 and 2.3 what figures use the methodology just described.**

Yes, you are correct. Since we removed the bootstrapping approach in the revised version of the manuscript (this includes the right panels in figures 2 to 5), we think that this should be less confusing now. To further clarify things, we added a short paragraph to section 3 that relates the left and right columns to the corresponding equations:

"The left columns in these figures show the emergent relationships including the uncertainty of the linear regressions (blue and orange shaded areas; see equation (5)) and the uncertainty in the observations (gray shaded area, see equation (7)). The right columns show the probability distributions of ECS in the original model ensemble (histogram) and the constrained distribution given by the emergent constraints (blue and orange line; see equation (8))."

**9. L156: how did you choose the 11 constraints you evaluate? Readers may think you cherry picked the constraints that behaved poorly if you don't say explicitly why you chose the ones you did.**

We choses these 11 emergent constraints based on their availability in the ESMVal-Tool. We added this to section 2.2:

"We chose these particular emergent constraints since these were already implemented in the ESMValTool (see section 2.4) at the time of writing this study, which greatly facilitated this analysis."

**10. L276 says Volodin was the first emergent constraint on ECS, which isn't true. Covey et al 2000 and Knutti et al 2006 provide earlier emergent constraints.**

We rephrased the sentence and removed the statement that Volodin (2008) was the first emergent constraint on ECS. Thank you for spotting this.

**11. L433: Bretherton and Caldwell (2020; https://journals.ametsoc.org/jcli/article/ 33/17/7413/348548/Combining-Emergent-Constraints-for-Climate) provide a multivariate technique for combining constraints on ECS. Doing so provided less conceptual insight than I expected -having most constraints predict high ECS led to the combined estimate also having high ECS with narrower uncertainty... which seems obvious in retrospect.**

Thank you for the interesting reference which we added to the summary section.

**1  Minor comments**

**1. L18 "which stem the major source" is wrong. I think you mean "which is the major source"?**

Changed in the manuscript.

**2.   You often say things like "the emergent-constrained best estimate". "Emergent-constrained" doesn't make sense. I think you mean the "emergent-constraint-constrained".**

Changed to "emergently-constrained" in the manuscript.

**3. L66: you already gave the range of ECS in the previous line, so saying CMIP6 models exceed 5K is redundant / unnecessary.**

Removed redundant part of the sentence.

**4. Eq 3: x should either include or exclude "_m" on \*both\* sides of the equation.**

We added the index $m$ to the function argument $x$ on the left side of equation (3).

**5. Eq 8 uses $P(y|x)$ from eq 6, which says it is an equation for $P(y|x_0)$. I think eq 6 is really true for all $x$ rather than just the observed value $x_0$. I suggest you remove mention of $x_0$ everywhere before eq 6.**

We replaced $x_0$ by $x$ everywhere except for equation (7).

**6. Sect 2.3: does it really take 100,000 samples to characterize uncertainty in a correlation between the 20-50 samples you're getting from the CMIP archive? I would guess 1000 iterations would be sufficient.**

We removed the bootstrapping testing in the revised version of the manuscript.

**7. L167: "Temperature (ERSST) is used": I'm confused because I thought you said you used HadISST on L164. Are you saying that the Brient + Schneider used ERSST?**

Yes, that is correct. We rephrased the sentence to clarify this.

**8. L351:'ve never seen the "(here: ...)" nomenclature you use. Do you mean "(e.g....)"?**

Replaced "here:" by "ZHA: ...; BRI: ..." in the first appearance and removed "here:" altogether in the second appearance.

---

## Author Comment (AC2) · 26 Sep 2020

Reviewer comments are marked in **bold**, our answers in red.

**Review of "Emergent constraints on Equilibrium climate sensitivity in CMIP5: do they hold for CMIP6?" by Manuel Schlund and co-authors.**

**In this study a series of mostly process-oriented emergent constraints that were developed on earlier model ensembles are applied to the latest CMIP6 ensemble. This is a very welcome attempt and in a broad sense testing scientific repro-**

[Figure]

**ducibility. My major concern is with the main conclusions drawn, or perhaps not drawn, from the results. The fact that estimated ECS based on these constraints increases roughly in proportion to the mean ECS increase from CMIP5 to CMIP6 suggests that these constraints are in not actually constraints on ECS, rather, at best they are constraints on the feedback processes they target. I develop argumentation this below, along with providing some more technical comments. I sign this review such that should the authors have any issues understanding my point they can contact me directly.**

**Sincerely,**

**Thorsten Mauritsen**

We thank the reviewer for the helpful and constructive comments. We have revised our manuscript in light of these and the other reviewer's comments we have received. A pointwise reply is given below.

**Major comments**

**Climate sensitivity is inversely proportional to the feedback parameter (lambda in equation 2) which in turn is a sum of a series of processes (sum of lambda_i). My take on the situation is that many of the emergent constraints (except COX) were successful on CMIP5 because inter-model spread in ECS in that ensemble was dominated by spread in low-level cloud feedbacks in the tropics. However, if any other feedback, e.g. water vapour feedback or any other cloud feedback, is biased in the ensemble as a whole then these kinds of process-oriented emergent constraints will necessarily be biased in their estimates of ECS. Probably even collectively since they thrive on the same kind of model spread, and so just because there are many studies that agree doesn't increase our confidence in**

their quantitative outcome. Likewise, if structural commonalities among models cause an unreasonable low inter model spread in some other feedback process then the emergent constraint is going to be over-confident. All in all, the results suggest that the original studies were overly confident and that changes in feedbacks not constrained by these studies cause them to be biased with a sign that cannot be determined (since CMIP6 probably also contains collectively biased feedback processes). Thus, these process-oriented emergent constraints are perhaps best thought of as constraints on the processes that they target, rather than constraints on ECS, and in extension the original studies have been disproven by the results of this study.

There are alternatives to process-oriented emergent constraints, though, one of them which is included in this study (COX, more about this study and why I think there is a shift below). Emergent constraints that use global temperature change as a predictor of ECS do not suffer from the same problem: even if one feedback is biased in a model ensemble the constraint can in principle still work since both global change and ECS are inversely proportional to the sum of feedbacks. Suggestions of emergent constraints of this kind include Last Glacial Maximum (Hargreaves et al. 2012 doi:10.1029/2012GL053872), Pliocene warming (Hargreaves and Annan, 2016 doi:10.5194/cp-12-1591-2016), and post-1970s warming (Jimenez-de-la-Cuesta and Mauritsen 2019 doi:10.1038/s41561-019-0463-y). All of these ideas have been tested across ensembles including CMIP6/PMIP4 (Tokarska et al. 2020 doi:10.1126/sciadv.aaz9549; Renoult et al. 2020 doi:10.5194/cp-2019-162), finding essentially unchanged results between ensembles. Other studies worth mentioning are Bender et al. (2010, doi:10.1007/s00382-010-0777-3) and Dessler and Forster(2018), although I haven't seen tests of these.

I think all of the above is rather straightforward and fairly easy to understand. I think the authors have everything at hand that they need to draw the conclu-

**sion that the process-oriented emergent constraints are not useful for estimating ECS, but rather should be better thought of as ideally constraining part of the cloud feedback. There are several places throughout that needs revising.**

We think the reviewer has a very good point. We therefore extended the section 4 adding the following discussion:

"Our findings suggest that the process-oriented emergent constraints (i.e. all of the emergent constraints investigated here except COX) are only successful in constraining ECS as long as the uncertainty in ECS is dominated by the same process or feedback. In the CMIP5 ensemble, cloud feedback is the main contributor to the spread in ECS with low-level clouds in tropical subsidence regions dominating the spread in cloud feedback (e.g. Ceppi et al. (2017)). If any other process or feedback is biased (or missing) in the ensemble as a whole, then these process-oriented emergent constraints will be biased in their estimates of ECS. The appearance of diverse new feedback processes in CMIP6 could explain reduced skill when applied to CMIP6 data, and a tendency for these to be positive would explain the upward shift in the model ECS distribution that is not captures by the CMIP5-trained constraints. Process-oriented emergent constraints are therefore perhaps best thought of as constraints on the processes that they target, rather than constraints on ECS.

Emergent constraints that use global temperature change as a way of constraining ECS could in principle not suffer from the same problem. If one feedback is biased in an ensemble the constraint might still work as both, global temperature change and ECS, might similarly reflect the sum of all feedbacks. Emergent constraints of this kind include e.g. the tropical temperature during the Last Glacial Maximum (Hargreaves et al., 2012), tropical temperature anomalies during the mid-Pliocene Warm Period (Hargreaves and Annan, 2016), and post-1970s warming (Jimenez-de-la-Cuesta and Mauritsen, 2019). This seems to be supported by the findings of Tokarska et al. (2020), who tested an emergent constraint for the transient climate response based on recent global warming trends on the CMIP5 and CMIP6 ensembles with similar results

for both model ensembles. However, these temperature-based estimates are sensitive to assumptions about forcings and unforced decadal temperature variations, which could also be systematically wrong, as could model-predicted relationships between feedbacks on short and long time scales that are implicit in most such measures. Indeed the significance of the COX constraint dropped as much from CMIP5 to CMIP6 ($p = 0.0032$ to $p = 0.1689$) as most of the other constraints in this study."

**I think it is not reasonable to provide best estimates of ECS in the abstract and summary based on this study for the following reasons:**

**1) The above issue.**

**2) Because the study does not apply the latest observations to the constraints, rather opts for using the original observations. This is a perfectly fine choice given the scope of the paper, but it does mean the constraints are not up to date.**

**3) Because the study uses an implicit flat prior which in case of weak data automatically leads to high-biased results.**

**I would instead suggest the authors cite percentage increases which is anyway all that is relevant here.**

We agree and therefore removed the best estimates of ECS from the abstract and the summary section and replaced them with percentage increases as suggested. The best estimates are now only given in the results section (3).

**Regarding the Cox et al. 2018 constraint, section 3.2, this is built on the Hasselmann (1976) single heat capacity model. In this model there is a linear relationship between Psi and ECS, however, and despite what they claim if you add a deep ocean to the model you obtain a non-linear relationship wherein the relationship is weaker for higher ECS, see Annan et al. (2020 doi:10.5194/esd-11-709-2020), their figure 5 (note flipped axes). If you look at how the CMIP6 models**

**are distributed they are simply situated in the flatter part of the expected curve, and if you fit a straight line to it you will obtain different slopes than for CMIP5.**

The reviewer has a good point. We therefore added the following sentence to section 3.2:

"For example, Annan et al. (2020) showed that the assumed linear relationship between $\Psi$ and ECS does not hold when adding a deep ocean to the model.".

**In this regard, and this applies not only to this study but most of these kinds, I am concerned with the general use of linear regression. The most silly example is SU constraint, where despite getting the wrong sign of the slope in CMIP6, you obtain a constraint on ECS. I think studies must be much more smart about their choice of statistical model, and not just use linear regression when non-linear behaviour is expected or other physical constraints can be applied such as a near-zero intercept, examples in Jimenez-de-la-cuesta-Otero and Mauritsen (2019), Annan et al. (2020) and Renoult et al. (2020). In case of process-oriented emergent constraints one could perhaps think of using Equation (2) in the form ECS~a/(b+x) where x is a process-oriented predictor. I am not saying the authors need to change this, but it would be worthwhile acknowledging that using linear regression, heedlessly, can lead to misleading and over-confident results.**

We added the limitations of the approach including the references that you mention to the methods section that introduces emergent constraint methodology (section 2.2) and also briefly refer to this in the summary section:

"A further limitation of our approach is the statistical model itself. Similar to many other emergent constraint studies, we use an ordinary least squares linear regression model for each emergent constraint. However, in some cases this might not be appropriate, e.g. when we expect non-linear behavior or when physical constraints can be used to derive further constraints for the regression model like a zero intercept (Annan et al.,
2020; Jimenez-de-la-Cuesta and Mauritsen, 2019; Renoult et al., 2020)."

**I found the discussion of statistical significance somewhat disturbing. The chosen thresholds seem purely subjective, as far as I can tell. I would suggest to delete this whole discussion which seem rather pretentious.**

Following this comment and a similar comment by Peter Caldwell (2nd reviewer) we decided to remove the whole bootstrap significance testing from the paper. Following the reviewers' suggestions, we replaced the bootstrapping method with a $t$-test on the correlation coefficient. The null hypothesis of this $t$-test is that no correlation exists between the predictor and ECS. In the revised version of the manuscript, we now give $p$-values of the emergent relationships that correspond to the probability that the absolute correlation is larger than $|r|$ even though the null hypothesis is true, i.e. the true underlying correlation coefficient is zero. Moreover, we do not use the $p$-values anymore to specify absolute significance (as you noted, our categories "highly significant", "barely significant", etc. were rather subjective), but only use them to specify relative significance, i.e. to indicate whether the statistical significance changes when moving from CMIP5 to CMIP6. Similar to our original bootstrapping approach, the new approach using the $t$-test shows that except for the ZHA constraint, all emergent relationships show a higher significance for the CMIP5 ensemble than for the CMIP6 ensemble.

**I found Sections 4 and 5 rather long and repetitive. I would suggest revising and sharpening.**

In order to address the two reviewers' comments we changed sections 4 and 5 substantially. Section 4 (discussion) now discusses possible reasons for the change in skill in the emergent constraints when moving from CMIP5 to CMIP6. Section 5 (summary) now gives a summary and discusses limitations of our study, some of which are now described in the methods section.

**Minor comments**

**19, 'of spread in ECS among models'**

Replaced "source of uncertainty in ECS" by "source of spread in ECS among models".

**36, 'concentration over pre-industrial levels'**

Added suggestion.

**69, I am not sure Forster et al. 2020 is correctly cited here**

We removed the reference to Forster et al. (2020).

**85-92, perhaps drop Delta from $F$, and when using a specific forcing in equation 2 write $F_{4x}$ or something?**

We greatly simplified section 2.1. In the revised version, equation (2) does not appear anymore.

**90-93. did you account for model energy leakage and drift? Concerning drift, some do account for this, but is not always obvious if there is a best way, nevertheless you must document what you did.**

We accounted for possible model drift when calculating ECS by subtracting a linear fit of the pre-industrial control simulation from the abrupt4xCO2 experiment. Other than that, no explicit corrections for drift or energy leakage have been done. We added this information to section 2.1 of the revised manuscript:

"In this calculation, the linear fit of a corresponding pre-industrial control run is subtracted from the 4xCO2 run to remove any model drift that is present in the control

climate without adding noise (Andrews et al., 2012). Other than that, we do not explicitly account for other problems such as energy leakage."

**120-123, there are also an intermediate option, e.g. the Cauchy prior used in Annan and Hargreaves (2011, Climatic Change). Regarding the uniform prior, please specify which cut-off you use.**

Following this comment and a similar comment by Peter Caldwell (2nd reviewer) we made the following changes to the manuscript:

"In this derivation of the probability $P(y)$ we do not assume any prior knowledge on ECS – in other words, that an ECS near 8 K would be deemed just as probable as one near 4 K if both are equally consistent with the observational best estimate $x_0$. We do this for simplicity. The PDFs would shift somewhat lower with a broad prior on processes instead (see Sherwood et al. (2020)), but we are concerned here with how outcomes compare using CMIP5 vs. CMIP6 data, rather than the exact ranges obtained. Such comparisons are not sensitive to the prior."

We emphasize that we focus on the relative differences in the constrained ECS distribution between CMIP5 and CMIP6, which does not depend on the choice of the prior.

**154-155, this is a misinterpretation, the IPCC 'likely' statements refer to 66-100 percent probability.**

Thanks for clarifying this! We replaced "IPCC likely" by "66% confidence interval (17–83%)".

**158-159, I felt this statement could be made more informative by explaining that it is the covariance of clouds with surface temperature anomalies.**

As suggested, we added this extra information to the beginning of sect 3.1.
**180, 'results to choices made in the analysis'**

Added suggestion to manuscript.

**236-263, these constraints seem to have some legacy with Fasullo and Trenberth (2012, Science), perhaps worth mentioning if the authors agree?**

We mentioned the Fasullo and Trenberth (2012) constraint in the sections describing the SU constraint (3.7) and the TIH constraint (3.8).

**276-277, it is incorrect that Volodin (2008) was the first emergent constraint on ECS, there is Covey et al. 2000 and Knutti et al. 2006 before then.**

We rewrote the sentence and removed the wrong part that stated that Volodin (2008) was the first emergent constraint on ECS. Thank you for spotting this.

**320, I would suggest deleting 'describing the real world'**

Removed "describing the real world".

**354-355, The idea and strength of an emergent constraint is that you use something you can observe to predict ECS. It really shouldn't matter if a process is slightly different in the warmer 2xCO2 world.**

We agree with the reviewer that it probably does not matter if a process is different in the future climate as long as the ESMs know about it. But there may be processes that are unimportant in the ESMs and hence not captured by the emergent constraints but that are important in reality. We clarified this by rephrasing the corresponding paragraph in section 4 of the revised manuscript:

"[...] While this assumption seems to make sense, we do not know whether the ESMs cover all relevant processes of the real Earth system. For example, it may be possi-

ble that there exist processes that are unimportant in the ESMs (and hence are not captured by the emergent constraints) but are actually important in reality. This lack of relevant processes may lead to an overconfident constraint. Thus, the more complex ESMs of the CMIP6 ensemble are more likely to capture relevant processes of the true climate which leads to weaker emergent relationships. On the other hand, emergent constraints on the less complex CMIP3 and CMIP5 ensemble may be overconfident."

**357-359, same applies here**

See answer above.

**360-361, this statement goes further than Zelinka et al. 2020. I would suggest replacing 'dominated by' -> 'to some extent associated with'**

Changed as requested by the reviewer.

**402-403, as per my above argumentation, I would be very careful with making this statement.**

Since we removed the bootstrapping testing from the paper, we also removed this whole paragraph.

**409, this might also have been shown by Klocke et al. (2011), check.**

In Klocke et al. (2011), they state: "This suggests that model weighting based on statistical relationships alone is unfounded and perhaps that climate model errors are still large enough that model weighting is not sensible.". In our opinion, this does not really fit into our argumentation, since we do not consider model interdependence and model skill at this point.

**421-426. the very same paper also shows that the ECS estimated from 4xCO2**

**runs is higher than twice that in 2xCO2 runs, and that the bias of the same order of magnitude.**

We did not find any reference to the reviewers point in Klocke et al. (2011) (https://doi.org/10.1175/2011JCLI4193.1).

---

## Author Response (AR2)

Reviewer comments are given in **bold**, our answers in red.

**The authors did a great job of addressing my concerns. I thought reviewer Thorsten Mauritsen brought up a good point about shift of constraints towards larger ECS indicating that most of these constraints are actually constraints on cloud feedback. The authors mention this in the paper, but could have redone their analysis with respect to cloud feedback rather than ECS in order to really explore this point. Nonetheless, no paper can address all questions and I think what the authors have is definitely sufficient for publication.**

We thank the reviewer again for the helpful and constructive comments that allowed us to substantially improve our manuscript during the previous round of reviews.

**Reply to Thorsten Mauritsen (Referee)**

Reviewer comments are marked in **bold**, our answers in red.

**Second review of "Emergent constraints on equilibrium climate sensitivity in CMIP5: do they hold for CMIP6?" but Manuel Schlund and co-authors.**

**Overall, I am happy with the revisions undertaken by the authors based on my first review. I have a few comments of minor nature that should be addressed prior to publication.**

**Sincerely,**

**Thorsten Mauritsen**

We thank the reviewer again for the helpful and constructive comments. We have revised our manuscript in the light of these. A point-by-point reply is given below.

Minor comments

**24, replace 'increased' -> 'widens'**

Changed in the manuscript.

**24-25, I think the text in parentheses is confusing, I recommend deleting**

Text in parentheses deleted.

**30-33, I am not sure I understand this text, I recommend revising. As it stands it seems to suggest that the emergent constraints that are process-based (those in this study except COX) are superior in constraining ECS to those that more directly constrain ECS using global change (Hargreaves 12+16, Jimenez 19, Tokarska 20, Renoult 20 etc.).**

We agree, the text was misleading. We changed this to

"Our results support previous studies concluding that emergent constraints should be based on an independently verifiable physical mechanism, and that process-based emergent constraints on ECS should rather be thought of as constraints for the process or feedback they are actually targeting."

in the revised version of the manuscript.

**94, Leakage is normally compensated for by subtracting imbalance from the control simulation. I think the authors do this.**

We adapted the text accordingly:

"In this calculation, the linear fit of a corresponding pre-industrial control run is subtracted from the 4xCO2 run to account for energy leakage and remove any model drift that is present in the control climate without adding noise (Andrews et al., 2012)."

**96-97, You might specify that these 6 percent is a model-based estimate of the compensation between changing feedback with time and the difference in ECS estimated by 2xCO2 and 4xCO2 (2x2 < 4).**

We added the following sentence:

"This number is a model-based estimate of the compensation between changing feedbacks with time and the differences introduced by considering a 4xCO2 run instead of a 2xCO2 run".

**144-145, I don't understand the meaning of this? Isn't the whole purpose of emergent constraints to utilise variations in model performance to better constrain something of interest?**

We agree. The sentence was removed.

**404-405, replace 'not suffer from the same' -> 'overcome this'**

Changed as suggested.

**411, a matter of tone, but I suggest replacing 'However' -> 'Nevertheless'**

Changed to "Nevertheless".

**413-414, In my first review I provided a possible explanation for this, i.e. that it is to be expected that models with higher ECS will yield this result. I am not sure that writing in this way provides the reader with an appropriate impression of the other studies mentioned in the paragraph.**

We agree. Since this paragraph focuses on emergent constraint that use the past warming trend, the mention of COX, which uses the temperature variability, might not be appropriate. We removed the sentence.

**432-433, see comments on 24-25.**

Removed the text in parentheses.

[revised manuscript text omitted]